

# Monitoring of total and off-road NO$_x$ emissions from Canadian oil sands surface mining using the Ozone Monitoring Instrument

Chris A. McLinden[1,2], Debora Griffin[1], Vitali Fioletov[1], Junhua Zhang[1], Enrico Dammers[3], Cristen Adams[4,*], Mallory Loria[1,†], Nicolay Krotkov[5], and Lok N. Lamsal[6]

[1]Air Quality Research Division, Environment and Climate Change Canada, Toronto, Ontario, Canada
[2]Department of Physics and Engineering Physics, University of Saskatchewan, Saskatoon, Saskatchewan, Canada
[3]Climate, Air and Sustainability (CAS), Netherlands Organisation for Applied Scientific Research (TNO), Utrecht, the Netherlands
[4]Resource Stewardship Division, Alberta Environment and Parks, Edmonton, Alberta, Canada
[5]Laboratory for Atmospheric Chemistry and Dynamics, NASA Goddard Space Flight Center, Greenbelt, MD, USA
[6]Universities Space Research Association, Columbia, MD, USA
[*]Now at: Air Quality Research Division, Environment and Climate Change Canada, Toronto, Ontario, Canada
[†]Now at: Astronomy Research Centre, University of Victoria, Victoria, British Columbia, Canada
**Correspondence:** Chris McLinden (chris.mclinden@ec.gc.ca)

**Abstract.**

The oil sands in Alberta, Canada is a significant source of air pollution. Observations from the Ozone Monitoring Instrument (OMI) on the NASA Aura satellite have been used to quantify NO$_x$ emissions from the surface mining region of the oil sands. Two related emissions methods were utilized, one for point and one for area sources, where OMI vertical columns densities of NO$_2$ were combined with winds from a meteorological reanalysis and a two-dimensional exponentially-modified Gaussian (EMG) plume model. This work better connects the two (point and area) emissions methods, discusses the interpretation of fit parameters, and the ability of OMI (and other sensors) to resolve emissions between neighbouring sources.

The two methods employed, in good agreement with each other, indicated an increase in emissions from about 55 to 80 kt[NO$_2$]/yr between 2005–2011, and flat thereafter. Reported emissions were typically 0-15% smaller, consistent to within uncertainties. In an extension of this methodology, OMI observations were combined with reported point source emissions to derive the more uncertain emissions component from the large off-road mining fleet. These were found to make up about 60% of total NO$_x$ emissions, also consistent with reported emissions. The OMI-derived 1.3%/year increase in fleet emissions and the 5.9%/year increase in bitumen mined, generally a good proxy for fleet emissions, can be reconciled by considering the evolution of the mine fleet over this period. OMI is therefore able to track the transition from US EPA Tier 1 standards, through Tier 4 standards, to the present, and in so doing demonstrates the efficacy of this policy. Furthermore, this analysis shows that had the fleet remained at Tier 1 this source would currently be emitting an additional 25 kt/yr.





## 1 Introduction

The utilization of satellite observations of atmospheric composition for estimating or constraining emissions has expanded significantly in the past two decades (Streets et al., 2013) following improvements in sensor spatial resolution and precision, retrieval algorithms, and emissions methodologies. While satellite observations alone can often be used as proxies for emissions, at best they only contain information on the emitted mass of a pollutant, and not the rate of emission. To bridge this divide additional information on the dispersion (via atmospheric transport) and removal (though physical and chemical processes, if relevant) must be incorporated.

One common approach to obtaining "top-down" emissions estimates is through the use of 3D chemical transport or air quality models that simulate all relevant physical and chemical processes. Model emissions are adjusted such that the model-simulated and satellite observed quantities match to within some tolerance. The specifics of the approach may depend on several factors such as resolution of the model and lifetime of the pollutant, and vary from the straightforward mass balance (e.g. Martin et al., 2003) to a more formal Bayesian approach in which a cost function is minimized (Streets et al., 2013).

An alternative class of top-down methods have become increasingly common over the past decade in which satellite observations are paired with wind information to directly estimate emissions of air pollutants such as $NO_x$, $SO_2$, $NH_3$, CO, and $CO_2$. This was done initially for point (or near-point) sources (Beirle et al., 2011; Pommier et al., 2013; Fioletov et al., 2015; Nassar et al., 2017; Dammers et al., 2019) but more recently for area sources (Fioletov et al., 2017; Beirle et al., 2019; McLinden et al., 2020; Fioletov et al., 2022). After several years of development, testing, refining, and validating, these methods are now being applied to emissions detection, verification and analysis (e.g. McLinden et al., 2016b; Ialongo et al., 2018; Goldberg et al., 2019; McLinden et al., 2020).

In this work, direct methodologies such as these have been applied to surface mining within the Athabasca Oil Sands Region (AOSR), located just north of the community of Fort McMurray (57°N, 111.6°W) ), in the Canadian province of Alberta (see Figure 1). The AOSR contains large deposits of bitumen, a viscous form of oil, which can be converted into a synthetic crude oil. In total, the AOSR is estimated to contain the equivalent of 170 billion barrels of oil, making it the second-largest reserve globally. In 2022, production from the AOSR was (the equivalent of) 3.3 million barrels of oil per day (mBPD) from bitumen, a number expected to rise to 4.0 mBPD by 2030 (Alberta Energy Regulator, 2021b).

Within the AOSR, about 20% of the proven reserves reside near the surface, at a depth of ∼75 m or less, and can be extracted through a process called surface mining. Surface mining first requires the removal of the overburden (muskeg and soil) in order to expose the deposit, followed by the extraction and transport of the raw bitumen to another on-site location for further processing to separate the bitumen from the sand and other impurities. For deeper deposits, in situ extraction methods must be employed in which steam is injected to reduce the bitumen viscosity so that it can be pumped to the surface. In either case, the bitumen is then converted into a synthetic crude oil through a process known as upgrading. Some facilities have on-site upgrading while others transport the bitumen elsewhere.

There are a number of environmental concerns associated with oil sands operations (e.g. Kelly et al., 2010), including water usage, deforestation (Rosa et al., 2017), greenhouse gas emissions (Rosa et al., 2017; Liggio et al., 2019; Wren et al., 2023),



and various potential environment effects to air, water, land, biota, and human health arising from the emission of pollutants. In particular, potential harmful effects on ecosystems from acidifying deposition (Makar et al., 2018), often referred to as cumulative effects. This refers to the environmental impact of past, present, and potentially future deposition, to the land and water. A comprehensive understanding of this impact, and how to mitigate it, requires an accurate knowledge of the emission rates of atmospheric pollutants (Galarneau et al., 2014; Gordon et al., 2015; Liggio et al., 2016; Li et al., 2017). Cumulative effects encompasses many pollutants, but in this work the focus are nitrogen oxides, $NO_x$ (or the sum of NO and $NO_2$) which are emitted as a by-product of combustion.

The Ozone Monitoring Instrument (OMI) satellite sensor has been used previously to help understand the distributions of $NO_2$ and $SO_2$ (McLinden et al., 2012, 2016a, 2020) in the AOSR. These studies showed that a "hot-spot", or local enhancement, in $NO_2$ can be readily observed over the surface mining. Likewise, OMI has been utilized to quantify $NO_x$ emissions with methods similar to that employed here from urban, industrial, and natural sources (Beirle et al., 2011; de Foy et al., 2014; Adams et al., 2019; Fioletov et al., 2022). This study builds on these to quantify $NO_x$ emissions from AOSR surface mining using the now 19 year time series from OMI with the specific aim of understanding how they have evolved given its continued expansion and the changes in vehicle emissions standards over time.

## 2 Measurements and Methods

### 2.1 Emissions Data

This study is focused on quantifying $NO_x$ emissions originating from a group of facilities withing a region in the northeast corner of the AOSR dominated by surface mining, defined here by a box bounded by latitudes of 56.80° and 57.46°N and longitudes of 112.0° and 110.7°W. This box (see Figures 1 and A1) contains all seven AOSR surface mining facilities, three relatively small in-situ facilities, as well as other minor $NO_x$ emitters such as the community of Fort McKay (pop. <1,000). Bottom-up inventories (Zhang et al., 2018) suggest that oil sands surface mining facilities and activities are responsible for roughly 95% of total $NO_x$ emissions within this area and so the term "surface mining region" is used here for convenience. Just south of this area is the community of Fort McMurray (pop. 70,000, 15+ km South), and further south still are numerous in-situ facilities (50+ km South).

Several sources of emissions information were used in this work. Annual emissions from point sources reported to the Canadian National Pollutant Release Inventory (NPRI) (Environment and Climate Change Canada, 2020) are available up to 2022. NPRI point source emissions are based on a variety of methods: some stacks employ continuous emissions monitoring systems (CEMS) (Alberta Government, 1998), others are based on engineering estimates, and some use a combination of both. The primary source of mine fleet emissions were created in accordance with the Environmental Protection and Enhancement Act (EPEA) (Alberta Environment and Parks, 2019) spanning 2010–2022, while those from the Cumulative Environmental Management Association (CEMA) for 2010 (Zhang et al., 2018) are also used for comparison. Related industrial activity data, including the total amount of oil sands mined on a monthly basis (Alberta Energy Regulator, 2021a), were helpful for interpretation and sampling corrections.



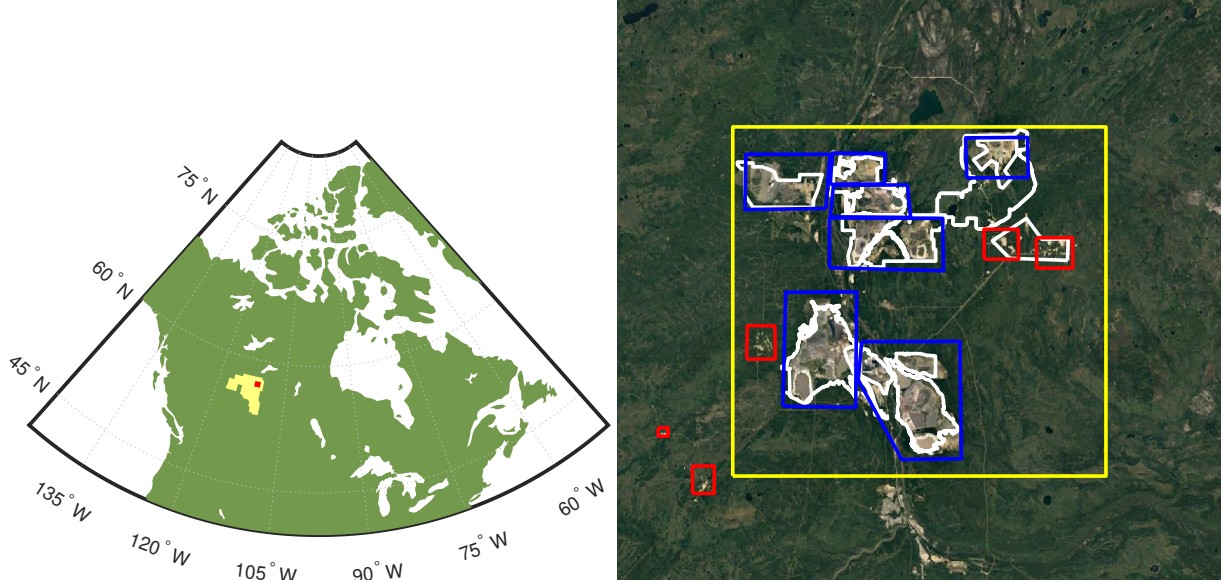

**Figure 1.** Left: Map of the Canada showing the Athabasca oil sands region (yellow) and the location of the surface mining area (red), which also corresponds to the area outlined by the yellow box in the right panel. Right: Map of the oil sands surface mining region. The yellow box indicates the region over which emissions are added and assumed to be surface mining, although there are some minor emissions from in-situ facilities that also reside within the yellow box. The blue boxes indicate surface mines, whereas the red boxes are smaller in-situ facilities. Map Data: Google, Image Landsat/Copernicius. © Google

Emissions of $NO_x$ from surface mining occur primarily from two source types. The first is the off-road vehicle fleet com-
85 posed of large, diesel shovels and trucks that operate 24/7. Three of the seven mining operations also have large point sources, stack emissions, from the upgraders that convert the bitumen into a synthetic crude oil. The in-situ facilities also have stack emissions although these are quite modest ($< 5\%$) in comparison. The databases indicate that, in 2010, within the surface mining region, roughly 38 kt[$NO_x$/yr] originated from the suface mining offroad fleet, 29 kt[$NO_x$/yr] from surface mine point sources, and about 1–2 kt[$NO_x$/yr] from in situ point sources. All other sources are estimated to emit considerably less than 1
90 kt[$NO_2$/yr] and are therefore ignored.

## 2.2 OMI Observations

OMI is a Dutch–Finnish instrument launched in 2004 on the NASA Aura satellite (Levelt et al., 2006, 2018). OMI measures back-scattered sunlight in the UV–visible using a two-dimensional detector that measures simultaneously at 60 across-track positions. Its spatial resolution at nadir is $13 \times 24$ km$^2$ but it gets progressively worse at larger off-nadir angles. A blockage
beginning in 2007 -– the so-called "row anomaly" -– has meant some track positions are no longer reliable (Levelt et al., 2018; van Geffen et al., 2019). As of 2020 more than half of the across-track positions are affected.



Version 4.0 of the OMI $NO_2$ standard product (Krotkov et al., 2018; Lamsal et al., 2021) is used here. It, like many UV-visible $NO_2$ retrievals, uses a three-step approach to determine tropospheric $NO_2$ vertical column densities (VCDs) from measured spectra, where the tropospheric VCD represents the vertically-integrated number density between the surface and the tropopause.

The first step in the retrieval is a determination of the total absorption by $NO_2$, quantified in terms of a slant column density (SCD). The SCD represents the $NO_2$ number density integrated along the path of the sunlight through the atmosphere. Since OMI measures scattered light the path is complex and includes one or more scattering events and/or surface reflections. SCDs are derived through an analysis of the measured spectra by exploiting the difference in absorption at nearby wavelengths and involves a spectral fit of reference spectra, one of which being the $NO_2$ absorption cross-sections, to the measured spectra . The second step is the removal of the stratospheric component of the total SCD. This is accomplished using a complex high-pass filtering approach. The final step is a conversion of the remaining tropospheric SCD into a tropospheric VCD via an air mass factor (AMF) that accounts for the sensitivity of the instrument to $NO_2$ for a particular scene, where VCD=SCD/AMF. In practical terms, a multiple-scattering model is used to calculate AMFs (Palmer et al., 2001) to account for the complex path of light through the atmosphere. AMFs depend on factors such as solar and viewing geometry, the presence of clouds, scene reflectivity, and the vertical distribution of $NO_2$.

AMFs are one of the largest sources of uncertainty in $NO_2$ VCDs (Lorente et al., 2017). In the OMI $NO_2$ standard product (SP, version 4.0), model profiles at 1° resolution are used for the vertical distribution. As these are coarser than the individual OMI pixels, this acts to smooth the VCD distribution. To improve the effective spatial resolution, AMFs were recalculated using higher resolution inputs and more recent oil sands emissions (McLinden et al., 2014). These inputs include $NO_2$ profiles from the Global Environmental Multi-scale – Modelling Air quality and CHemistry (GEMMACH), discussed in section 2.3, and an annually-varying monthly-mean MODIS clear-sky reflectivity at 0.05° (Schaaf et al., 2002) smoothed to 15 km. Furthermore, snow pixels are identified using the Interactive Multisensor Snow and Ice Mapping System (Helfrich et al., 2007) which was found to be the snow product best suited for distinguishing between snow-free and snow-covered scenes (Cooper et al., 2018). When snow was detected, AMFs were then calculated using a MODIS-derived 15 km snow reflectivity as per McLinden et al. (2014).

OMI $NO_2$ VCDs were filtered as follows: track positions affected by the row anomaly are not used, and the track positions at the edge of the detector, 1–7 and 54–60 which correspond to the coarsest spatial resolution, are also removed. Only observations made for solar zenith angles of 75° or less are retained. At a latitude of 57°N this removes the majority of data for the months of November to January. Note that a seasonal correction was developed to account for this in the determination of annual emissions (see section 2.4.4, below).

## 2.3 The GEM-MACH Air Quality Forecast Model

This study utilizes output from the Canadian Global Environmental Multi-scale - Modelling air quality and Chemistry (GEM-MACH) regional operational air quality forecast model (Moran et al., 2010; Pavlovic et al., 2016; Makar et al., 2017; Pendlebury et al., 2018) which covers most of North America. GEM-MACH is an on-line chemical transport module that is embedded



within the ECCC weather forecast model. GEM-MACH was utilized as described in McLinden et al. (2014) where monthly-mean $NO_2$ profiles at 15 km $\times$ 15 km were computed for use in the AMF calculations. As discussed below in section 2.4.3, GEM-MACH was also used to better quantify the $NO_2/NO_x$ ratio.

## 2.4 Satellite-derived Emissions

Emissions are derived using a direct approach in which wind information is paired with individual OMI $NO_2$ VCD observations. In this step, vertical profiles of the $u-$ and $v-$components of wind speed are obtained from ECMWF (European Centre for Medium-Range Weather Forecasts) reanalyses (pre-September 2019: ERA-Interim (Dee et al., 2011), September 2019 and after: ERA-5).

   From here, two related emission algorithms are employed: one developed for isolated, near-point sources (Fioletov et al.,
2015) and one for multiple or area sources (Fioletov et al., 2017). Both algorithms rely on a two-dimensional exponentially-modified Gaussian (EMG) plume function which translates the emissions into a spatial distribution. The functional form for the 2D EMG, provided in appendix B, is essentially a modified version of the traditional Gaussian plume with simple (e-fold) chemistry (Stockie, 2011) but is integrated in the vertical dimension and accounts for the finite size of the source and spatial resolution of the instrument. The EMG plume distribution varies with upwind/downwind and across-wind distance from a
reference point that represents the location of the source, windspeed, and also depends on three parameters: pollutant mass, $m$, its effective lifetime $\tau$, and a plume width parameter, $\sigma$. The emission rate, assuming a mass balance, is simply $E = m/\tau$. Henceforth the EMG will be described as a function of $(E, \tau, \sigma)$ for simplicity.

   The effective lifetime describes the rate of decay of the pollutant due to chemical (e.g., conversion into other species) or physical (e.g., dry deposition) loss. It is well known that the lifetime of $NO_x$ depends on $NO_x$-itself, and a recent study
(Laughner and Cohen, 2019) elucidates this in the context of theoretical $NO_x$-lifetime curves for various volatile organic compound reactivity regimes. Previous studies using EMG-like functions found effective lifetimes of 2–5 hours by following the downwind decay of $NO_2$ VCDs (Beirle et al., 2011; de Foy et al., 2015; Lange et al., 2021).

   The effective plume width (or spread) parameter, $\sigma$, is interpreted as a combination of a plume diffusion parameter, $\sigma_{\text{diff}}$, the spatial resolution of the satellite instrument, $\sigma_{\text{pixel}}$, and the physical size or extent or size of the source itself, $\sigma_{\text{size}}$. Given
the Gaussian nature of the EMG, it is reasonable to assume that these are related through a functional form resembling,

$$\sigma^2 = \sigma_{\text{diff}}^2 + \sigma_{\text{pixel}}^2 + \sigma_{\text{size}}^2. \tag{1}$$

Considering the physical extent of the oil sands, $\sigma_{\text{size}}$ is expected to be tens of km. Similarly, given the OMI spatial resolution, $\sigma_{\text{pixel}}$ should be on the order of 20 km. The remaining term, from equation (B4), is $\sigma_{\text{diff}}^2 = \alpha \cdot y^\beta$ where $y$ is the downwind distance, and $\alpha = 1.5$ km and $\beta = 1$ (Fioletov et al., 2015). Conventional Gaussian plume models use values more like $\alpha = 0.33$
160  km and $\beta = 0.86$ (Stockie, 2011). The larger value for $\alpha$ accounts for additional uncertainty in the wind downwind of the source. In the case of OMI, with its rather coarse resolution, decreasing $\alpha$ has virtually no impact on the fit.

   The precise relationship between $\sigma$, $\sigma_{\text{diff}}$, $\sigma_{\text{size}}$, and $\sigma_{\text{pixel}}$ was explored in appendix F using a simple model in which a collection of true-Gaussian plume point sources were used to simulate a near point source (of some radius $\sigma_{\text{size}}$) and smoothed





to a specified satellite resolution (for pixel size $\sigma_{\text{pixel}}$). Fitting this combined distribution to an EMG allowed a relationship
between these various terms to be established. Figure F1 shows $\sigma$ has a dependence reminiscent of $\sigma_{\text{size}}^2$ and $\sigma_{\text{pixel}}^2$, and for
small values of both, values less than 1 km which suggests $\sigma_{\text{diff}}$ is small. Figure F1 will be used below to help interpret results.
This further suggests that only for instruments with substantially higher spatial resolution than OMI, and likely even TROPOMI
or TEMPO (Zoogman et al., 2017), would this $\sigma_{\text{diff}}$ term need to be refined.

The EMG plume function is used in two ways. The first is an inverse mode where combined OMI-wind data are fit in
order to derive ($E, \tau, \sigma$). The second way is in a forward mode to predict the spatial distribution of VCDs (at an OMI-like
spatial resolution) by prescribing winds and values for ($E, \tau, \sigma$). The input emissions used for this forward mode can be those
derived from OMI (in the inverse mode), in which case the intent is to 'reconstruct' the original OMI observations to test for
consistency. Alternatively, a modified or even completely independent source of emissions, $E$, can be used, with $\tau$ and $\sigma$ as
derived from OMI, in order to examine the predicted spatial distribution of VCD (at OMI spatial resolution).

### 2.4.1 Point Source Method

The first emissions approach considered is the point source method of Fioletov et al. (2015), used previously to derive emissions
of $SO_2$ (Fioletov et al., 2016; McLinden et al., 2020), $NH_3$ (Dammers et al., 2019), and $NO_x$ (Griffin et al., 2021). Observations
from many OMI overpasses are combined using a rotation procedure in which all observations are re-positioned about a
reference location according to their individual wind directions such that, after rotation, all observations share a common wind
direction (Pommier et al., 2013). In other words, the upwind/downwind and cross-wind positions, relative to the reference
location, of each observation is preserved. In this way multiple overpasses can be analyzed as a single, average plume. In
this method values of ($E, \tau, \sigma$) are determined that minimize the difference between the EMG plume function and the OMI
observations in the rotated plume. Since $\tau$ and $\sigma$ are non-linear parameters, a non-linear solver is used. A variation of this
method is also used in which $\tau$ and $\sigma$ are prescribed, thereby allowing $E$ to be determined using a linear regression which is
more stable and often means fewer observations are required. Details on the practical implementation of this method are given
in appendix C.

### 2.4.2 Multi-Source Method

The second method, developed for multiple or area sources (Fioletov et al., 2017, 2022), was also employed. This is a com-
plimentary approach to the point source method in that the same EMG plume function is used except here ($\tau, \sigma$) are specified
and emissions from multiple locations are solved for simultaneously. At each location an EMG basis function is generated for
all OMI observations being used in the fit. The OMI VCDs are modelled as the sum of all EMG functions which represent the
local sources, and a background term, which represent the $NO_2$ that would be present in the absence of local sources. The back-
ground can be a simple constant offset, or allowed to vary linearly in latitude and longitude. A multi-linear regression is then
performed which provides a set of emissions that minimize the difference between the OMI and modelled (or reconstructed)
VCDs. This fit is performed in the original latitude-longitude frame (ie, not in the rotated frame) and a positive constraint is
placed on the emissions.





In this multi-source method, the choice of emission locations is arbitrary. If the emission sources are well known then these locations can be used. Likewise, a grid of locations can be used when their spatial distribution is uncertain or complex. One aspect of this approach is that $\tau$ and $\sigma$ must be specified. While the effective lifetime can be estimated from a model, there is a question as to how precisely to do this. An alternative approach, and the one adopted here, is to use the effective lifetime derived from the point source method. The choice of $\sigma$ is not necessarily obvious since it encompasses multiple parameters, as indicated by equation 1. It should be chosen such that $\sigma_{\text{size}}$ represents the physical size of the emissions. For a collection of point sources, then $\sigma_{\text{size}} = 0$. For a grid representing an area source, $\sigma_{\text{size}}$ should be on par with the grid spacing. Of course, this assumes $\sigma_{\text{OMI}}$, or at least $\sigma_{\text{diff}}^2 + \sigma_{\text{OMI}}^2$, is known. One method to estimate these other terms is to fit $\sigma$ for a true point source as is discussed below. Additional information is provided in appendix D.

### 2.4.3 Other Considerations

One important parameter that must be specified is the altitude of the winds, where the altitude is chosen to represent the average height of the $NO_2$ plumes. As emissions are roughly proportional to wind speed, and wind speeds typically increase rapidly with height through the boundary layer, this also represents one of the larger sources of uncertainties. Previous studies used winds averaged over the lowest 500 m for urban, primarily vehicle (Goldberg et al., 2019), emissions while another study used winds over the lowest 1 km (Fioletov et al., 2015).

In this work, wind heights are based on the observations from aircraft campaigns conducted in 2013 and 2018 which studied pollutant transformation downwind of the oil sands (Li et al., 2017; Liggio et al., 2019). Observations of of $NO_x$ within plumes were found to reside at altitudes between the lowest altitudes sampled by the aircraft (100–300 m above the surface) and 800 m. In these same plumes, $SO_2$, which originates only from the upgrader stacks (e.g. McLinden et al., 2020), was found at 400–800 m. This indicates that $NO_x$ from upgraders is best represented at 400–800 m, while emissions from the fleet at 0–400 m. On this basis, and assuming roughly equal emissions from the stacks and the vehicles as suggested by the bottom-up inventory, winds, averaged between the surface and 800 m will be used for the majority of the analysis herein. Furthermore, an uncertainty of 100 m is assigned to the wind height, which translates into a difference in wind speed of roughly 10% since, in the vicinity of the the oil sands, wind speeds on average were found to roughly double between the surface and 1000 m. For situations where fleet and stack emissions are treated separately, winds averaged over 0–400 and 400–800 m are used, respectively.

As OMI only observes $NO_2$ a correction must be applied to account for the missing NO. Typically this is done by estimating the $NO_x/NO_2$ concentration ratio and scaling the derived emissions by this. Previous studies often used more generic values of 1.3–1.4 (Beirle et al., 2011; Goldberg et al., 2019). Here, a site-specific value was determined from the GEM-MACH model. The annual average surface mining concentration ratio for $NO_x/NO_2$, considering grid-boxes within a 40 km radius and using the same monthly sampling pattern as OMI, is 1.50. The larger value is due in part to the inclusion of non-summertime chemistry where the photolysis of $NO_2$ is typically slower, particularly at higher latitudes.

However, using a single concentration ratio does not account for any spatial variability. Close to the actual site of the emissions this ratio can be larger as $NO_x$ is primarily emitted as NO (relatively less $NO_x$ is $NO_2$) while further downwind the opposite is true (relatively more $NO_x$ present as $NO_2$). This acts to skew the shape of the (rotated) plume thereby impacting the



**Table 1.** OMI NO$_x$ emissions uncertainty budget when considering three years of observations.

| Error Category | Source | Uncertainty | |
|---|---|---|---|
| | | Magnitude (%) | Type (R=Random, S=Systematic) |
| Air Mass Factor | all systematic sources | 14 | S |
| Data Filtering | Remove snow pixels | 7 | S |
| | ±3 track positions | 1 | S |
| | cloud fraction ±0.1 | 3 | S |
| EMG emissions fit | Fitting errors | 5–10 | R |
| | Fit downwind distance | 3 | S |
| | Fit box Width | 4 | S |
| | Different Methods | 3 | S |
| Winds | Wind speed and direction | 6 | R |
| | Wind height | 10 | S |
| NO$_x$/NO$_2$ | NO$_x$/NO$_2$ ratio | 8 | S |
| TOTAL | | 20–25 | % |

fit. Similar to Griffin et al. (2021), the GEM-MACH model is used to quantify the impact of this: VCDs of NO$_2$ and NO$_x$ for a year of simulations over the oil sands, along with model winds, are each fit to the EMG plume function using the point source method. Taking the ratio of NO$_x$ to NO$_2$ emissions derived in this way reveals a ratio of 1.63, which is roughly 10% higher than the simple concentration ratio. This 10% difference is a systematic effect that should be largely independent of location

provided emissions are from combustion. This value of 1.63 was used here. See appendix F for additional information.

### 2.4.4   Uncertainties

There are several sources of uncertainty that limit the accuracy and precision of these OMI-derived emissions. Table 1 shows a breakdown, with each listed as a random or systematic sources of uncertainty, for the point source method. They are arranged under five categories – AMF, data filtering, the EMG fit, winds, and the NO$_x$/NO$_2$ ratio.

Only systematic errors in AMFs are considered since random errors should largely cancel as multiple years of data are analyzed together. A previous study (McLinden et al., 2014) found systematic uncertainties to be 14% due to approximations with how aerosols are handled and the use of a Lambertian surface albedo. Data filtering refers to what OMI data are included or excluded in the analysis for which different thresholds could be argued: exclusion of snow pixels, track position cut-off, which really denotes the maximum size of the pixels, and maximum cloud fraction. These are systematic effects that collectively

represent an 8% uncertainty.





The EMG fit contributes a random uncertainty to emissions related to the quality of the fit itself, which varies from 5–10% depending on the amount of data used. Beyond that, there are small systematic effects when it comes to the spatial domain used in the fit itself, specifically the cross-wind, upwind and downwind distances. There is also a small uncertainty included here that reflects how different variants of the algorithm produce slightly different emissions.

From Section 2.4.3, a systematic error of 10% from using an incorrect wind height is reasonable. Also, random errors in wind speed and direction were estimated to be 6% from a previous study (McLinden et al., 2016b). The $NO_x/NO_2$ emission ratio used here is 1.63 as discussed in section 2.4.3. An 8% systematic uncertainty is assigned to this category, arrived at by considering the differences between the average of the aircraft measurements from Figure E2 and multiple years of GEM-MACH output.

There are also potential sources of bias due to the sampling of the OMI instrument: overpasses are only in the early afternoon, and there is an unequal distribution of observations throughout the year. The first of these is only an issue if there is a diurnal cycle in emissions. Hourly emissions of $SO_2$, co-emitted with $NO_x$, from the upgraders indicate no consistent difference in time of day emissions. Likewise, the heavy hauler fleet of trucks are used 24/7 and it is assumed there is no systematic diurnal variation in these emissions. High-resolution GEM-MACH forecasts assume flat $NO_x$ emissions for both source types (Zhang 260 et al., 2018). In addition, the fitting algorithm itself smooths the diurnal cycle impact to some extent as $NO_x$ emitted before the overpass time is sampled downwind.

Any potential seasonality bias is largely eliminated using OMI observations from all months. Nonetheless, the SZA cutoff of $75°$ and the change in cloud cover with season means not all months contribute equally. The impact of this is assessed using monthly bitumen mining data (Alberta Energy Regulator, 2021a). Each OMI pixel is assigned a bitumen-mined value 265 equal to its corresponding monthly total. An annual average bitumen-mined value was then calculated averaging over all pixel values, thereby capturing the OMI sampling, and these were then compared to the average assuming an equal weighting. The resultant bias is typically about 1–2%, with one year reaching 5%. This correction was applied directly to the OMI emissions. See appendix F for the time series and additional information.

Constructing an error budget for the multi-source method is more challenging, but it is argued here that it would be very 270 similar since the largest contributors are systematic errors from the AMFs and winds, which would be common to both. As discussed below, fitted parameters from the point source method are used as inputs to the multi-source which also speaks to their connection. Given all this, and what was found to be a high degree of consistency in emissions as obtained from each method, this uncertainty budget is used for both approaches.

It is worth noting that the uncertainties derived here are smaller than that reported in other satellite-based $NO_x$ emission 275 studies. There are several reasons for this: (i) location-specific and temporally-resolved fitting parameters ($\tau$ and $\sigma$) , (ii) the use of multiple years of data to reduce fit uncertainties, (iii) aircraft observations of real plumes providing more confidence in wind heights, and (iv) a location-specific and more detailed consideration of the $NO_2/NO_x$ ratio.



## 3 NO$_x$ Emissions from the Oil Sands

### 3.1 Surface Mining as a Point Source

An initial assessment of NO$_x$ emissions for oil sands surface mining is made using the point source method. Here, a reference location of 55.06°N, 111.55°W is used as it is the location of the maximum in the mean OMI NO$_2$ VCD distribution, as shown in Figure 2a. Winds are taken as the average between the surface and 800 m (see section 2.4.3). Figure 2 shows a summary of the emissions procedure considering the entire OMI time series (2005–2022). These maps were created using a 2×2 km$^2$ grid and a 12 km oversampling radius (Fioletov et al., 2011). Panel (a) shows the mean NO$_2$ VCD along with an outline of the

surface mines and the reference location. Panel (b) uses the same data as (a) but averaged in the rotated frame, as a function of downwind and cross-wind distance from the reference location. In this frame the reference location corresponds to (0,0). There is a clear plume-like feature that peaks just downwind of the reference location before it decays close to background some 100–150 km downwind. The peak value in the rotated frame is roughly 10–15% larger as compared to the original frame as a result of a greater cohesion in the distribution. Panel (c) shows the map that results from the non-linear fit of the individual VCD

observations to the EMG plume function in the rotated frame, and it can be seem to capture the shape of the average plume in (b). The fit parameters are E= 69.7 ± 2.2 kt[NO$_2$]/yr, $\tau = 3.0 \pm 0.1$ hours, and $\sigma = 20.6 \pm 0.3$ km, where the uncertainties are statistical only, an indication of how well the EMG represents the OMI/wind observations. The overall uncertainty in the emissions would be roughly ±20%, from section 2.4.4. The final panel, (d), shows the reconstructed VCD distribution as found by rotating the fitted values in panel (c) back to the original co-ordinates. This generally resembles the mean OMI VCD map in

panel (a) but has a smaller peak value and appears more symmetric, consequences of the reconstruction having all emissions, effectively, released from a single point.

   This approach is repeated, but now considering successive three-year time periods, where three years are combined to help ensure the stability of the non-linear, fitted parameters $\tau$ and $\sigma$, particularly in latter years when the row anomaly means less than half of all pixels are usable. The fitted parameters are presented first, deferring any discussion of the emissions to

section 2.4. Figure 3a shows the time series of the mass of the NO$_2$ enhancement. This represents the average mass of NO$_2$ that is present as a consequence of the surface mining NO$_x$ emissions discussed in section 2.4. It is seen to increase with time, peaking mid-time-series ($\sim$2013), and them decreasing slightly. The dip in 2016 is attributed to the large forest fire just south of the surface mining in and around the community of Fort McMurray (Adams et al., 2019) that significantly impacted production throughout the month of May. Note that NO$_x$ from the fires themselves originated 20 km south of the surface mines

and two weeks later approached the southernmost edge (MNP (Meyers Norris Penny), 2017). There are only one or two days where it is conceivable that there could be some misidentification of fire-NO$_x$ for oil sands but given its proximity and wind direction, these data tended to be filtered due to higher cloud fraction. There is no obvious sign that COVID-19 public health restrictions affected emissions in the most recent years, consistent with the 2020–2022 production data remaining roughly constant compared with previous years (Alberta Environment and Parks, 2019). The increase in the 1-sigma error bars (in all

panels, a–c) reflects the decrease in the number of OMI pixels due to the onset and expansion of the row anomaly.







**Figure 2.** Summary of the point source emissions procedure using 2005–2022 OMI NO$_2$ VCDs. (a) Mean OMI VCD over the surface mining region of the oil sands. The black lines denote the different mining operations; (b) mean OMI VCD after rotation showing plume-like structure; (c) Reconstructed spatial distribution using EMG plume and fitted parameters; (d) Fit to rotated VCD using 2D EMG plume function, with fitted parameters of E=69.7 kT/yr, $\tau = 3.0$ hours, and $\sigma = 20.6$ km. The black triangle in the panels (a) and (c) show the reference location used in the emissions retrieval and corresponds to the (0, 0) point in panels (b) and (d).

For comparison, the three-year running average of in-situ NO$_2$ is also shown in Figure 3a. Here, the average from the Fort McKay monitoring station, located between the northern and southern cluster of mines, is used but subtracted from it is the NO$_2$ from a background station at Fort Chipewyan, roughly 200 km to the North. This difference in station values is used as the



**Figure 3.** OMI point source emissions and parameter time series: (a) mass of the NO$_2$ enhancement and, for comparison, the difference in surface volume mixing ratio between a surface mining and background site (see text), (b) effective lifetime of plume, and (c) plume width parameter and average OMI pixel size. All OMI-derived quantities are based on three-years of observations, such that, e.g., 2006 considered observations from 2005–2007, with the exception of 2005 which is based on two-years, 2005–2006.

mass reflects an enhancement in NO$_2$ above background levels due to the local sources. While this simple comparison cannot
be considered a validation, their general consistency provides confidence in the approach.

From Figure 3b, the effective lifetime displays a modest variation with time, between 2.7–3.2 hours, peaking mid-way through the time series. It is well known than NO$_2$ can impact the abundance of OH, and hence its own lifetime (Valin et al., 2013), and it is in this context that this is explored further, below.

The variation of $\sigma$ is shown in Figure 3c, where it is seen to increase from $\sim 18$ km in the mid-2000s to $\sim 21$ km by 2009
and with a smaller rate of increase to $\sim 24$ km near the end of the OMI record. This is broadly consistent with an expansion of





the surface mining activities in the north, including new mining operations coming online, which effectively increase the area of $NO_x$ emissions. Also shown is the average effective size of the pixels used in the analysis, calculated as the square-root of pixel area, and it is seen to increase slightly with time due to the row anomaly. While similar to the increase in $\sigma$, the magnitude of the increase in pixel size is much smaller. The link between pixel size and $\sigma$ is discussed below. From Figure F1, 21 km

pixels for a surface mining radius of 30 km, would predict a value for $\sigma$ of about 21 km, consistent with that derived here.

These results can be contrasted with a previous study (McLinden et al., 2012) in which, between 2005–2011, the mass was seen to increase at a similar rate over this period but was a factor of 2–3 smaller. This is due primarily to the differences in AMF, where the former study used the VCDs as provided in the KNMI version 2 data product (Boersma et al., 2011). In that data product, the model simulated $NO_2$ profiles were for background conditions as it has no emissions in the AOSR (McLinden

et al., 2014). Other differences are due to the data version and the more sophisticated fitting in this work. It is this difference in fitting that also complicates any quantitative comparison between the fitted width parameters in these two studies.

In order to examine the variation in lifetime more thoroughly, an effective-VCD was derived using the mass from 3a (converted to molecules) and divided by the square of the width parameter from 3c. This would peak mid-time-series and be a minimum towards the early and later years. Note that such a quantity is not the mean VCD, but simple one that accounts for the

changing spatial extent of the source with time. Plotting this effective lifetime vs effective VCD shows a reasonably compact relationship, as shown in Figure 4, with a correlation coefficient of 0.74. Note that the correlation degrades to 0.62 when mass, as opposed to effective VCD, is considered. For pure chemical loss of $NO_x$ via $NO_2$+OH, $[OH] = 1/(k_{OH+NO_2} \cdot \tau)$ can be used to estimate [OH], giving values between 6–8$\times 10^6$ molecules/cm$^3$, also shown in Figure 4. For the 2013 flight campaign in the oil sands, summertime OH was estimated at $10 \times 10^6$ molecules/cm$^3$ (Liggio et al., 2016). The 2013 value from this OMI

analysis, 6$\times 10^6$ molecules/cm$^3$, is generally consistent since it considers all seasons (albeit weighted towards spring and summer) and wintertime values would be considerably smaller. This analysis suggests that at least some of the temporal variation of $\tau$ is real, and highlights the importance of using the period-specific $tau$ when estimating emissions. Using the 2005–2022 mean $\tau$ could lead to a 10–15% additional error and skew the trend.

### 3.2 Surface Mining as a collection of Point Sources

When quantifying emissions from a location where the spatial scale of the emissions is larger than $\sigma$ (as is the case here), the multi-source method may be more appropriate than the point source method. Here a grid of potential source locations covering the entire surface mining region is used. Initially an 8$\times$8 km$^2$ grid is defined over an area of 250 km $\times$ 250 km centered on the surface mining. Each grid-box is treated as a (potential) point source, analogous to the approach used in Fioletov et al. (2017). As discussed in section 2.4 and appendix D, emissions are derived for each grid-box such that their combined VCDs match the

OMI observations. The positive constraint ensures most of these fitted values, and hence emissions, are zero.

In this approach $\sigma$ and $\tau$ must be specified. While the effective lifetime, $\tau$, can be taken from the point source analysis, $\sigma$ is not as straightforward. The reason for this is because $\sigma$, as derived above, has embedded within it the spatial scale of the entire surface mining source (that is, the $\sigma_{rmsize}$ term in equation 1). Following the discussion in section 2.4, additional insight into how OMI views a point-source was obtained by analysing a different location, one which is more of a true point source. For this





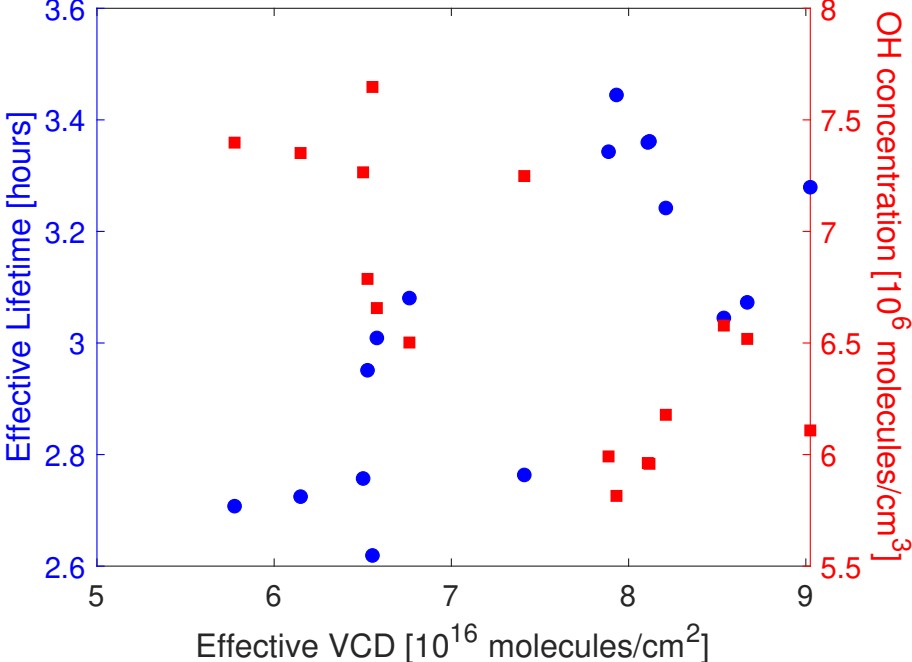

**Figure 4.** Relationship between effective VCD and lifetime and inferred OH concentration. Each point represents one three-year value from Figure 3.

purpose, the Poplar River Power Station (NPRI ID 2079), a coal-burning power plant located in southernmost Saskatchewan (49.05°N, 105.49°W), was considered. Average reported emissions over the 2005–2020 period are 13.4 kt[$NO_2$]/yr, where OMI finds 11.5 kt[$NO_2$]/yr, suggesting a reliable fit. From this analysis a $\sigma$ of 11 km was derived. This can be compared with the value from Figure F1 which, for $\sigma_{\mathrm{pixel}} = 21$ km and $\sigma_{\mathrm{size}} = 0$, yields a value of about 13 km, which is generally consistent. A value of $\sigma_{\mathrm{pixel}} = 11$ km is used for the multi-source method.

Initially all OMI observations spanning 2005–2022 are analyzed together and so a value of $\tau = 3.0$ hours (see Figure 2) is used. Performing the multi-linear fit provides a grid of retrieved $NO_x$ emissions. A comparison of the mean VCD map with the reconstruction using these retrieved gridded emissions is given in Figure 5a–b. The multi-source reconstruction now closely resembles the observations, and is much more realistic than the point source reconstruction from Figure 2d. The map of retrieved, gridded emissions used for Figure 5b is shown in Figure 6a. The emissions map shows large values over the

surface mines, as expected, with the largest corresponding to the upgrader at the Syncrude-Mildred Lake and Suncor facilities. Total surface mining emissions, calculated by summing grid-boxes within the rectangle shown, are 63.5 kt/yr. Outside of this box, other small but non-zero emissions can be seen. Some of these correspond to known $NO_x$ sources such as the city of Fort McMurray, in-situ mining operations, and other small non-oil and gas related sources. Some of the small but non-zero emissions retrieved are likely the result of systematic biases in the VCDs.





**Figure 5.** Summary of the multi source emissions retrieval using 2005-2020 OMI NO$_2$ VCDs. (a) Mean OMI VCD over the surface mining region of the oil sands. The black lines denote the different mining operations (this panel is the same as Figure 2a); (b) Reconstructed spatial distribution using the gridded emissions from Figure 6b. Panels (c) and (d) are the VCDs reconstructed using emissions from the NPRI point source database and OMI-derived area fleet emissions, also from Figure 6b. Note that the background is omitted from panels (b) and (c).

This retrieval grid is 31 x 31, totalling 961 grid-boxes. An important issue is the degree of independence of each box. One would not expect OMI, with its ∼21 km pixels, to be able to resolve emissions on an 8×8 km$^2$ grid. The multi-source method is based on EMG plume functions which serve as the basis functions in the multi-linear regression, and the extent to which they are correlated can be used to help determine the ability of OMI to spatially resolve emissions. At a distance of 8 km,




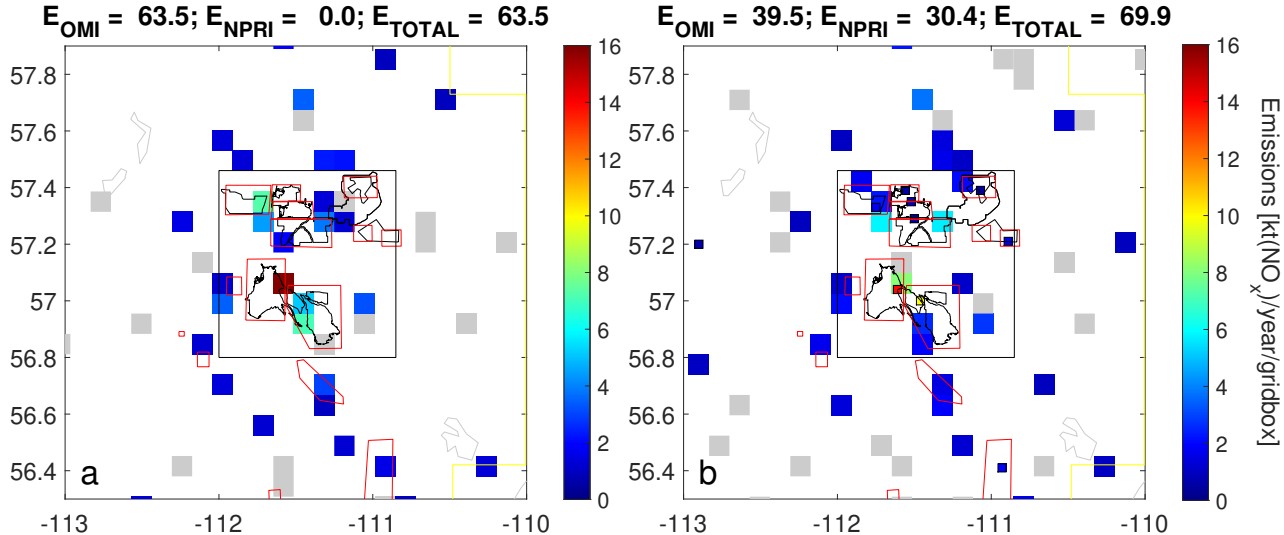

**Figure 6.** (a) OMI-derived NO$_x$ emissions retrieved on an $8\times8$ km$^2$ grid using the multi-source method. These emissions represent the total of point and area sources. (b) as (a) but when point source emissions from the NPRI database, totalling 30.6 kt/yr, are specified. These gridded emissions represent area emissions only. Gray boxes represent retrieved emissions between 0 and 0.5 kt/yr. In (a) the total emissions within the box are 63.5 kt/yr; in (b) the total is 69.9 kt/yr, where 30.4 kt/yr were specified and the remaining 39.5 kt/yr were derived from the multi-source method.

neighbouring OMI NO$_x$ plume functions have a correlation of 0.92. Given uncertainties in the data and approximations of the method, this confirms that OMI is unable to disentangle them. A more reasonable correlation threshold is 0.5, below which individual sources can be resolved. For OMI NO$_x$, it was found that sources must be separated by about 22 km. This is explored further in appendix F, where Figure F2 shows how achieving this 0.5 threshold separation distance, $z_{\min}$, varies with $\sigma$, and to a lesser extent, $\tau$.

The minimum distance required to distinguish between neighbouring emissions concept is very important for satellite emissions monitoring. An expression relating $z_{\min}$ directly to satellite resolution, $\sigma_{\text{pixel}}$, can be estimated by combining Figures F1 and F2. The result is shown in Figure 7. Estimates for several satellite data products are shown. While longer-lived species tend to require a slightly larger separation, the main driver is spatial resolution. This figure suggests TROPOMI and TEMPO (Zoogman et al., 2017) are able to resolve NO$_x$ emissions sources that are ~9 and 7 km apart, respectively. This approach does not account for real data characteristics, source strength (and relative strength of the two sources), and quantity of data used. One might also argue that a correlation coefficient threshold of 0.5 is too stringent, and that perhaps 0.7 might be more appropriate.

The spatial scale of the individual surface mines is 5–20 km but they often are separated by little or no distance. It is on this basis, albeit with one exception in appendix D where this is explored further, that OMI emissions derived from the



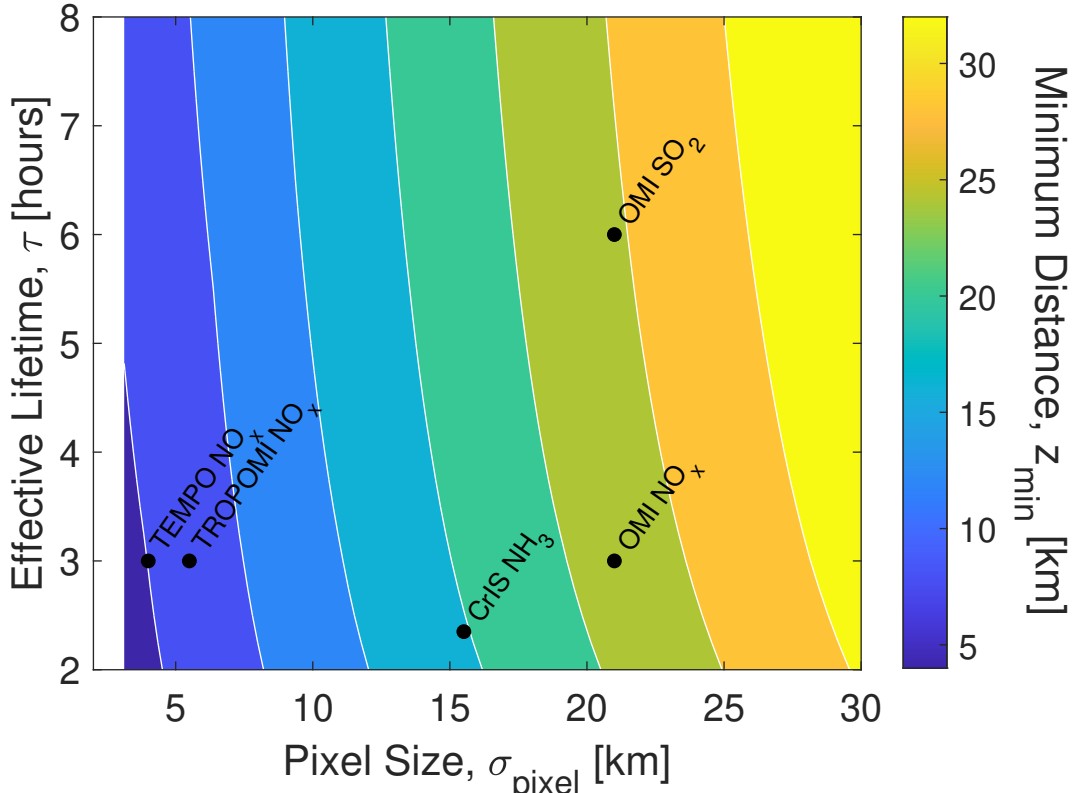

**Figure 7.** Minimum distance required to distinguish between two point sources as a function of satellite pixel resolution and effective lifetime. Values for several satellite emissions data products are also denoted.

multi-source method were simply summed over the surface-mining box. It is noted that TROPOMI, with its superior spatial
resolution, should be able to separate out emissions for the majority of these mines.

    An alternative implementation of the multi source method was used to derive vehicle fleet emissions. Recall that $NO_x$ is primarily emitted by point sources, or stacks, and the off-road vehicle fleet. Some of the stack emissions make use of CEMS, and overall stack emissions are believed to be well known. Therefore, if the contribution to the OMI observations from the stacks can be removed, the remainder, from the fleet, can be fit using the same approach as above. This requires an initial
step in which VCDs are reconstructed for each OMI observation, assuming stack emissions from NPRI, with these then subtracted from the actual OMI VCDs. This approach also allows an additional refinement as the height of the winds used for the prescribed stack emissions (400–800 m) can be different from those used to derive the off-road fleet emissions (0–400 m).

    In practice this was done by first taking the NPRI emissions in the region, averaged over 2005–2020, and using these to reconstruct VCDs as shown in Figure 5c. The peak over the southern mines reflects the larger emissions from the two
upgraders separated by about 10 km. These VCDs were then subtracted from the OMI observations, and the remainder used in the multi-source retrieval. The reconstructed total VCDs, those from the multi-source added to the NPRI VCDs from Figure



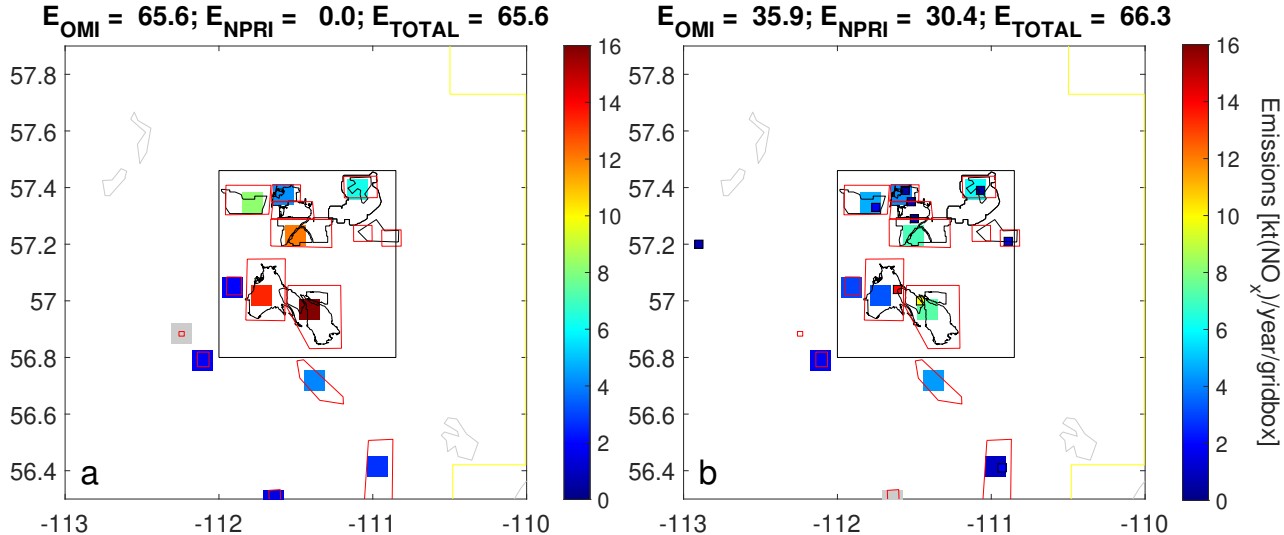

**Figure 8.** (a) OMI-derived $NO_x$ emissions retrieved using the multi-source method where emissions locations are assigned to the centre of mining facilitues. These emissions represent the total of point and area sources. (b) as (a) but when point source emissions from the NPRI database, totalling 30.6 kt/yr, are specified. In (a) the total emissions within the box are 65.6 kt/yr; in (b) the total is 66.3 kt/yr, where 30.4 kt/yr were specified and the remaining 35.9 kt/yr were derived from the multi-source method.

5c, are virtually identical to the that from Figure 5b and so it is not shown. Figure 5d shows the VCD reconstructed considering only the retrieved emissions which are attributed to the mine fleet. This VCD distribution are more homogeneous through the area, reflecting the more diffuse nature of this source.

The corresponding emissions map is shown in Figure 6b, where the small squares represent the prescribed NPRI point source emissions, and the grid box values are the retrieved (fleet) emissions. Total (point+fleet) emissions for the two methods differ due primarily to how the wind height is handled. Had the same winds been used in both, then the totals would be the same (to within 0.1–0.2 kt/yr) and one could have obtained the same result by simply subtracting the NPRI total from that in Figure 5a.

     Two more variants of this method are considered. Here, instead of a grid, potential source locations are limited to known
emissions sites, including industry and the city of Fort McMurray. The resultant emissions maps are shown in Figure 8. The squares reflect the emissions retrieved from each location. No attempt was made to account for the spatial scale of the mining operation; rather each was treated as a point source. As can be seen, the emissions maps are similar overall to the grid approach in both spatial distribution and surface mining total.

### 3.3    Emissions Time Series

As with the point source method, emissions using the multi-source approach were also derived considering three-year time increments and using the effective lifetime time series from Figure 3b. Maps of average VCD maps and their reconstructions are shown in Figures D1 and D2, respectively. The emissions time series is shown in Figure 9.



Several variations of the multi-source were utilized, analogous to what was discussed in section 3.2. A summary of these are given in Table E1. Considering these together with the point source results shows relatively little variation amongst them, typically a few percent with the largest being 15% for a single 3–year periods. Based on this and the difficulty in determining which method is 'best' it was decided to take the mean of several methods, according to Table E1), as the final emissions value, with their variability taken as a measure of uncertainty.

Emissions are found to steadily increase over the OMI record from about 55 to 80 kt/yr as seen in Figure 9a between 2005 and 2010, remain roughly flat thereafter. Comparing total $NO_x$ emissions, there is a good overall agreement between OMI and the bottom-up estimates. The bottom-up values are consistently 0–15 kt/yr smaller, but within the OMI uncertainties. Fleet emissions, as derived by prescribing stack emissions, are shown in Figure 9b. These generally follow the total emissions since the stack emissions only display a modest increase with time. The magnitude of the fleet emissions uncertainties are taken to be the same as that from the total emissions in Figure 9a. Fleet emissions were found to comprise about 60% of the total emissions. Neither reported nor satellite emissions show much variation over the 2010–2022 period for which both are available.

## 4 Discussion

As can be seen in Figure 10a, the average rate of increase in fleet emissions is 1.3%/year since 2005, with the bulk of the increase occurring 2005–2011. Yet the rate of increase in the mass of oil sands mined, which is generally considered a reasonable proxy for fleet emissions since they must transport the bitumen from the mine to the separation facility, is considerably larger. This apparent discrepancy may be reconciled by considering the evolving standards as the Canadian vehicle $NO_x$ emissions transitioned from unregulated (Tier 0) to present-day standards (Tier 4i/4) (Environment Protection Agency, 2016). As part of an agreement with the US, Canada adopted the EPA Tier 1 standard for heavy-duty, non-road vehicles (9.2 g[$NO_x$]/kWh) in 2000, the Tier 2 standard (6.4 g[$NO_x$]/kWh) in 2006, and then the Tier 4 standard (3.5 g[$NO_x$]/kWh) in 2012. There were no Tier 3 standards for this class of vehicle. Lower tier trucks could still be used as these stricter regulations were phased in, but any new or replacement engines were required to comply with the standard of the time. With a typical engine lifetime of 12 years, the benefits of Tier 4 regulations will not be fully realized until the late-2020s (M.J. Bradley & Associates, 2008).

An emission intensity metric is defined here as the mass of $NO_x$ emitted from the mining fleet per unit mass of mined oil sands, and this is shown in Figure 10b. It is seen to decrease at 3.8%/year. To contrast this with an expected decline, the composition of the vehicle fleet is required. Unfortunately, it was only in 2018 as part of the new Alberta Emissions Inventory Reporting Program (AEIR) that industry was required to report on fleet composition. While useful as a snapshot, this means there is no direct information on fleet evolution over the study period.

One alternative is a study delivered to Environment and Climate Change Canada in 2006 projecting the make-up of the mine fleet going forward (M.J. Bradley & Associates, 2008). Their projected mine fleet composition is shown in Figure 11. This study predicted the largest growth between 2005 and 2010 where total vehicles increased from about 200 to 500, before leveling off at 600. This is the reason for the rapid increase in Tier 2 fraction. For comparison the AEIR data indicate a fleet size of 730 large (> 750 hp) trucks in 2018. A comparison with AEIR reported tier fractions is also shown in Figure 11, where the





**Figure 9.** Comparison of reported and OMI-derived NO$_x$ emissions for (a) total and (b) fleet in the oil sands surface mining region. Thick error bars represent the random uncertainty and thin error bars represent total (random and systematic) uncertainty. The shading indicates the variability among the various methods for deriving emissions (and is included in the total uncertainty).

weighting was done using total vehicles as a function of tier as well as fuel consumed by tier. There is general agreement with the projections, with tier 2 vehicles composing the majority. One difference is the fraction of tier 1 vs. tier 4, with there being fewer tier 4 vehicles than projected. Note that one large facility in the AEIR did not report which tier their vehicles belonged to and so these trucks were excluded from this analysis.

As a second alternative, a simple model was constructed beginning with 205 Tier 1 trucks in 2005. Trucks were increased at the same rate as total bitumen mined, with a fraction of existing trucks being replaced each year with ones at the current tier. Assuming a 12 year lifetime (M.J. Bradley & Associates, 2008), where for simplicity lifetime is taken as an *e*-fold such that 8.0% of trucks (or engines) were replaced each year. As suggested by the bitumen mined time series in Figure 10a, the





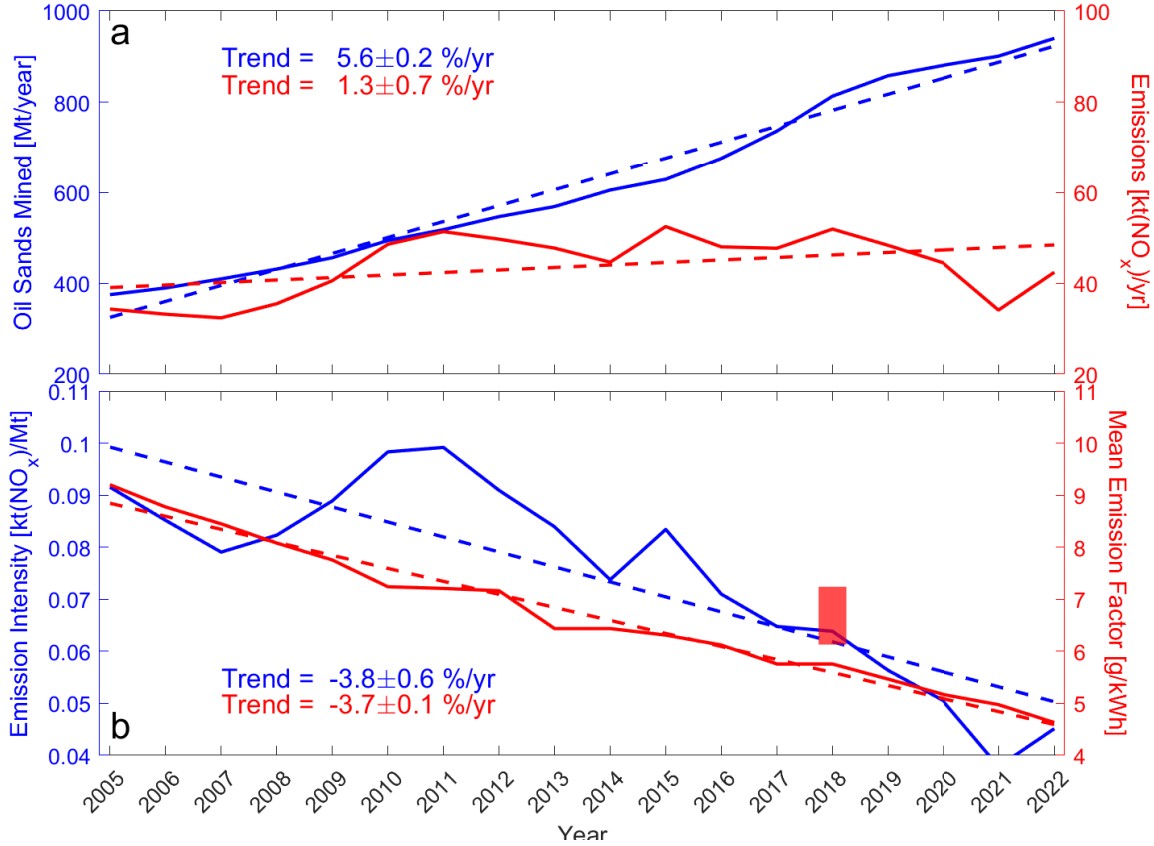

**Figure 10.** (a) Time series of mass of oil sands mined and fleet emissions from OMI. Dashed lines are linear trend lines, with the calculated trends indicated. (b) Emission intensity (defined as the ratio of fleet emissions to oilsands mined) and an estimated mean emission factor (see text). The shaded red bar show an estimate of the mean emission factor using reported fleet data.

total trucks increased steadily and doubled over the timeframe considered. The largest difference between this approach and
the M.J. Bradley & Associates (2008) and AEIR data being the difference in Tier 2 truck fraction.

While the 730 large (> 750 hP) diesel trucks from the 2018 AEIR mine fleet report make up less than half of the 1610 total vehicles listed in the report, they accounted for 92% of the fuel consumed and so would also be responsible for the large majority of $NO_x$ emissions. Overall, the AEIR total number of trucks, and the breakdown among the tiers, agree better with the projections. Nonetheless, the model remains a useful sensitivity study.

Assuming each truck uses the same amount of fuel and emits according to its standard, the weighted average emission factor can be computed for the projected fleet compositions. This was done by weighting the Tier 1/2/4 emission standard (9.2/6.4/3.5 g[$NO_x$]/kWh) with the fractions from Figure 11. Thus 2005 would have a value of 9.2 since it is entirely Tier 1, and 2006 would be slightly smaller since it is about 85% Tier 1 and 15% Tier 2. This metric, from Figure 10b, changes at −3.7%/year, a decrease that is essentially the same as that for the emissions intensity. In 2018, the emission intensity as defined



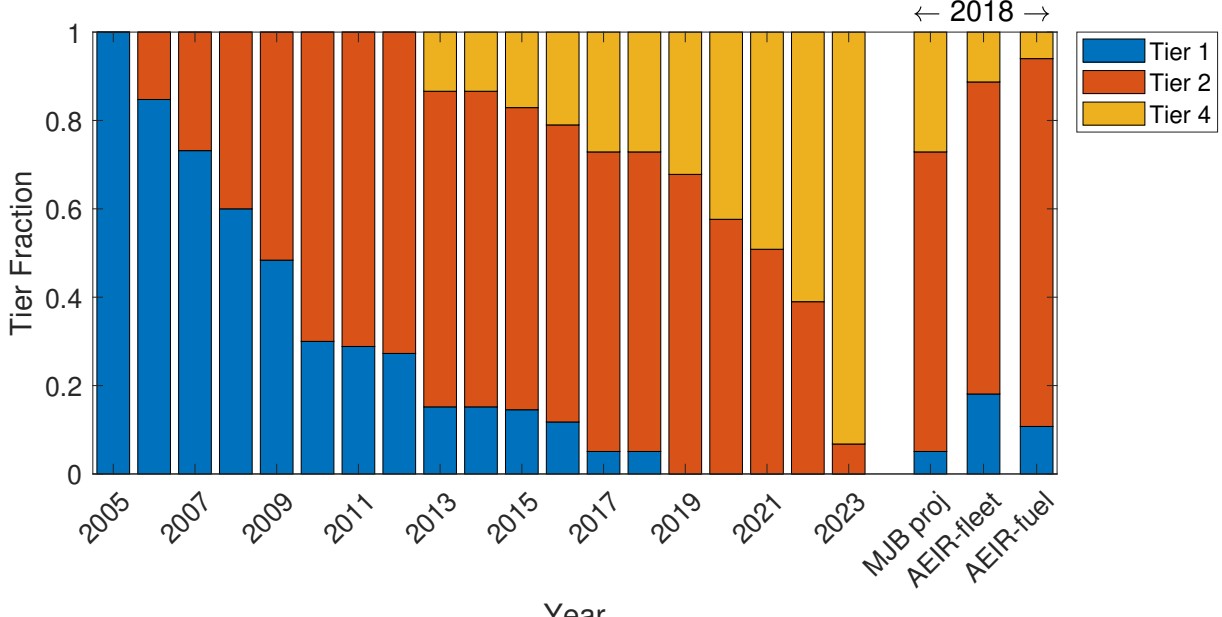

**Figure 11.** Composition of mine fleet according to emissions tier, shown as a cumulative fraction. The time series shows projections from the M.J. Bradley & Associates (2008) study for large ($> 750$ hp), diesel vehicles. The three on the right compare 2018 projected values with the 2018 AEIR fleet report, total vehicles and fuel consumed.

above has declined by about 31% since 2005, and the mean emission factor using the fleeted projections has declined by 46%. These can be compared with the change in AEIR mean emission factor, assuming an initial value of 9.2 g/kWh, the Tier 1 $NO_x$ standard. A decline of 21–31% is calculated, with the range depending on whether total vehicles or total fuel consumed was used to weight the mean.

Figure 10b suggests that the efficacy of the EPA emissions standards, at least in the case of the oil sands fleet, adopted
by Canada are effective at or close to the level expected. Considering Figure 10b further, had emission intensities remaining constant at 2005 levels, with 2005 being the final year of Tier 1 standards, and increased with oil sands mined (5.6%/yr) instead of that observed (1.3%/yr), the fleet emissions reduction brought about from the adoption of Tier 2 and later Tier 4 standards, relative to Tier 1 is roughly 25 kt[$NO_2$]/yr in 2022. A value of 25 kt[$NO_2$]/yr is comparable to that emitted from urban area (minus local industry) of roughly 3M people using an approximate value for a per-capita rate of emissions of 7 kt[$NO_2$]/yr/(1M
people) derived from (Fioletov et al., 2022). Over the past 18 years, roughly 210 kt[$NO_2$] less $NO_x$ has been emitted as a result.

The multi-source method can also be used to examine emission scenarios. For example, the impact on the average $NO_2$ VCD distribution due to a 20% reduction in fleet emissions or the opening of a new operation emitting 15 kt[$NO_x$]/yr could readily be evaluated. This is demonstrated in Figure 12 by comparing $NO_2$ VCD maps calculated using 2021–2023 OMI-derived fleet and NPRI emissions to VCDs derived using these fleet emissions increased by 50% as might be expected for Tier 1 vehicles.
In order words, Figure 12b represents the scenario avoided by implementing Tier 2 and later Tier 4 standards.



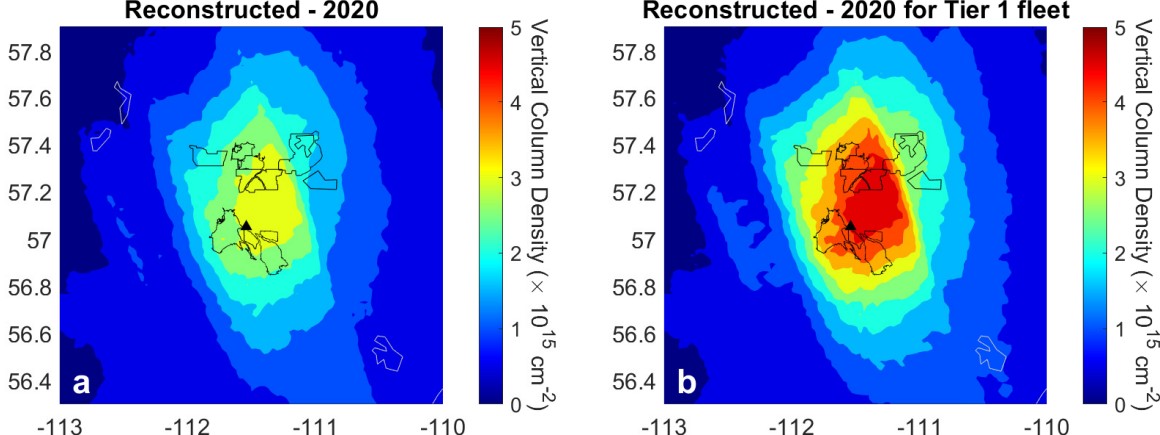

**Figure 12.** (a) Reconstructed 2020 NO$_2$ VCD map using OMI-derived fleet emissions and NPRI point source emissions. (b) As (a) but with fleet emissions increased by 50% to approximate expected VCD distribution assuming Tier 1 emission standards. Note the colourbar range here is different as compared with previous figures.

## 5 Summary and Conclusions

OMI observations of NO$_2$ tropospheric VCD were used to quantify NO$_x$ emissions from the surface mining region of the Canadian oil sands between 2005 and 2022. Two different, though related, emissions algorithms were found to give very similar results. One assumes all emissions emanate from a single (near) point source, whereas the other treats the area as multiple point sources, or an area sources. These methods both utilize wind speed and direction from a meteorological reanalysis and a two-dimensional exponentially-modified Gaussian (EMG) plume model.

OMI-derived emissions were found to increase from 55 to 80 kt[NO$_2$]/yr between 2005–2011, and then remain roughly flat afterwards. These were found to be roughly 0–15% larger than reported emissions, but given a 20% uncertainty in the OMI emissions, this difference is not significant. In an additional variation of this methodology, OMI observations were combined with reported point source emissions to derive the more uncertain emissions component from the large off-road (heavy-hauler) mining fleet. These were found to make up about 60% of total NO$_x$ emissions, consistent with about 55% from reported emissions. The 1.3%/year increase from this source and the 5.9%/year increase in bitumen mined, generally a good proxy for fleet emissions, can be reconciled by considering the evolution of the mine fleet over this period. OMI is therefore able to track the transition from US EPA Tier 2 standards (in 2006) through to Tier 4 (in 2012) to the present and in so doing demonstrate the efficacy of this policy. Furthermore, this analysis shows that had the fleet remained at Tier 1 (emission intensity) this source would be emitting an additional 25 kt/yr, or roughly 30% of the current total, an amount equivalent to that from a city of $\sim$3M inhabitants (Fioletov et al., 2022).



Some improvements over other studies include a more consistent, emissions-based scaling to convert $NO_2$ to $NO_x$, which was found to give a conversion factor about 10% larger than simple $NO_x/NO_2$ concentration ratio. Another improvement lay

in the use of an evolving effective lifetime, reflecting the fact that $NO_2$ impacts its own loss rate. In addition to emissions, this work better connects the point source and area source emissions methods and discusses the interpretation of fitting parameters and the ability to resolve emissions from sources in close proximity.

*Data availability.* Original OMI Level 2, orbit-based $NO_2$ data are available from the NASA Goddard Earth Sciences Data and Information Services Center (https://aura.gesdisc.eosdis.nasa.gov/data/Aura_OMI_Level2/OMNO2.003/). ECMWF (European Centre for Medium-

Range Weather Forecasts) reanalysis data (ERA-Interim and ERA5) are available here, https://www.ecmwf.int/en/forecasts/datasets/browse-reanalysis-datasets. The North-America-wide OMI $NO_2$ VCDs reprocessed using ECCC air mass factors used in this work are available at https://collaboration. cmc.ec.gc.ca/cmc/arqi/OilSands_satellite_NO2data in netcdf format.

*Code and data availability.* The analysis code was written in Matlab, and is available from the authors upon request.

### Appendix A: The Oil Sands Surface Mining Facilities

The individual mining facilities are shown in Figure A1.

### Appendix B: The Two-dimensional Exponentially-Modified Gaussian Plume Function

The two-dimensional exponentially-modified Gaussian (EMG) is used to model the vertical column density, VCD, distribution of the plume as seen from a satellite instrument. It is preferred over the traditional Gaussian plume (Stockie, 2011) as it better accounts for the finite spatial extent of the source itself, and the relatively coarse spatial resolution of the satellite

observations being utilized. Mathematically it is the convolution of a two dimensional Gaussian (i.e., integrated through the vertical dimensional) and an exponential. It depends on the cross-wind distance, $x$, and the downwind distance, $y$, each in km and relative to the location of the emission source, and the wind speed, $s$, in km/hr and where the wind is aligned in the $y$-direction. Carrying out the convolution leads to the following functional form, described by equations B1-B5,

$$\text{VCD}(x,y,s) = a \cdot \text{EMG}(x,y,s) = a \cdot f(x,y) \cdot g(y,s) \tag{B1}$$

such that $a$ is a constant and EMG is the plume function based on the function $f$ and $g$, where

$$f(x,y) = \frac{1}{\sigma_1\sqrt{2\pi}} \cdot \exp\left(-\frac{x^2}{2\sigma_1^2}\right) \tag{B2}$$

$$g(y,s) = \frac{\lambda_1}{2} \cdot \exp\left(\frac{\lambda_1(\lambda_1\sigma^2 - 2y)}{2}\right) \cdot \text{erfc}\left(\frac{\lambda_1\sigma^2 - y}{\sqrt{2}\sigma}\right) \tag{B3}$$





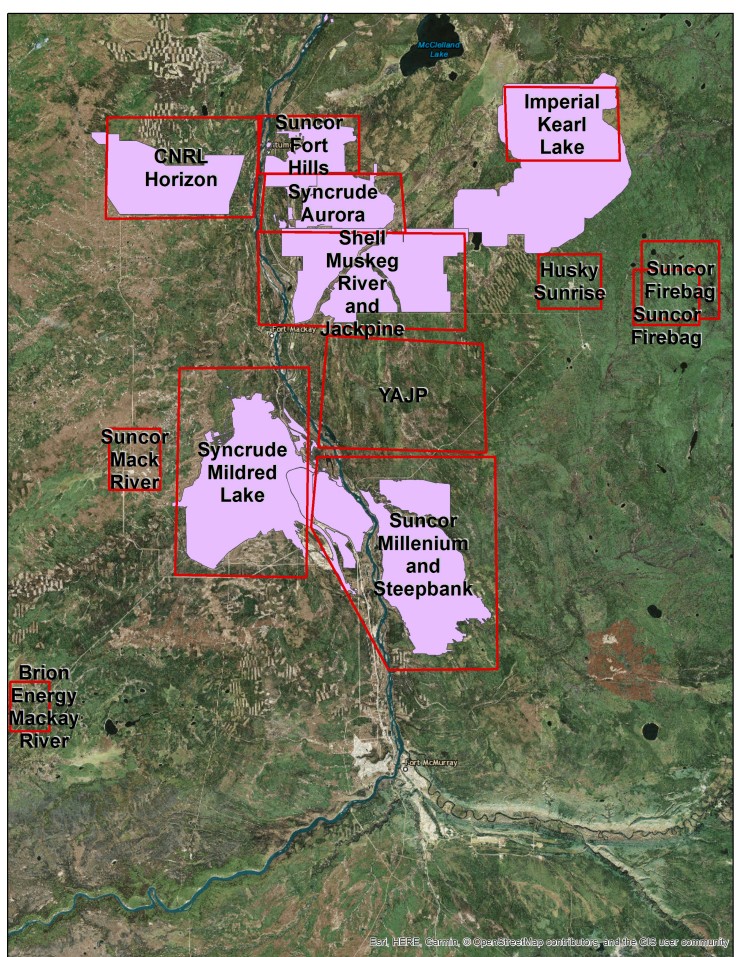

**Figure A1.** Location and name of all mining facilities.

where

$$\sigma_1 = \begin{cases} \sqrt{\sigma^2 + 1.5y} & , \ y > 0 \\ \sigma & , \ y \leq 0 \end{cases} \tag{B4}$$

$$\lambda_1 = \frac{1}{\tau \cdot s} \tag{B5}$$

and $a$ is a factor representing the concentration enhancement by the source and is directly proportional to the mass of the enhancement, $m$, $\sigma$ is the parameter describing the width of the Gaussian [in km], $\tau$ is the effective lifetime (in hours) and $\text{erfc}(x) = \frac{2}{\sqrt{\pi}} \int_x^\infty e^{-t^2} dt$. This formulation is the same as in Fioletov et al. (2015) except that $y$ has been replaced by $-y$ so

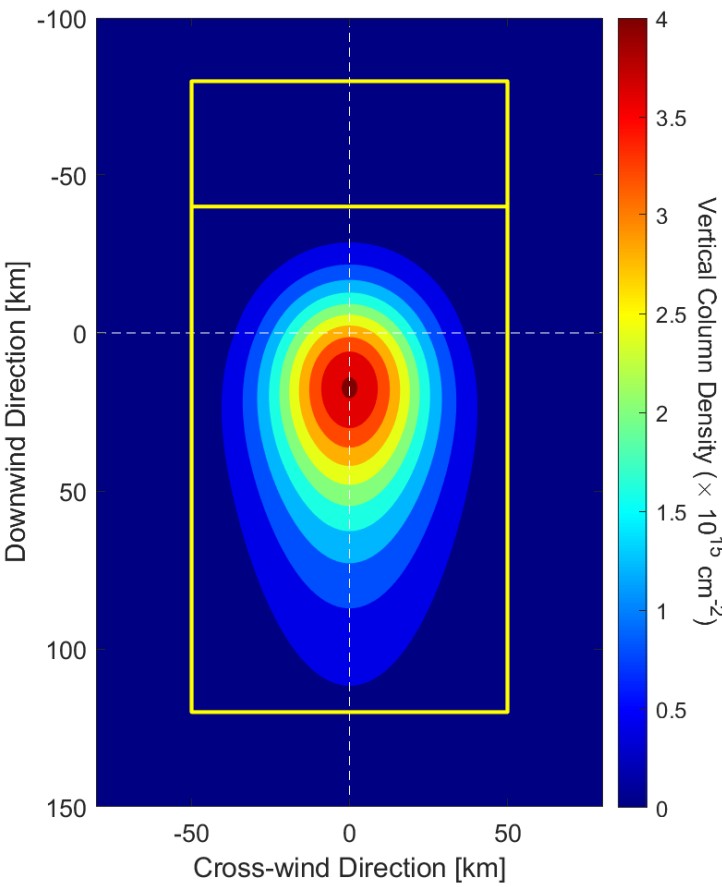

**Figure B1.** Sample of NO₂ VCDs generated using the EMG plume function as defined in equations B1–B5. In this example $E$=50 kt[NO$_x$]/yr, $\tau = 2.5$ hours, $\sigma = 18$ km and a constant wind speed of 15 km/hr were specified. The upper box shows the domain used to calculate the background, where the lower box shows the domain over with the EMG function is fit. The limits in $x$ are $\pm 50$ km and $y = -80$ to $-40$ km for the background, and $y = -40$ to 120 km for the fit. No background NO₂ was added in this example.

that distances downwind of the source are now positive. The emission rate is given by $E = m/\tau$, and ultimately the distribution is characterized by $(E, \tau, \sigma)$.

Equation B2 can be recognized as the standard Gaussian describing the cross-wind distribution. In equation B3, $g(y, s)$, is a convolution of a Gaussian and and exponential in the downwind direction, where the exponential accounts for first-order loss due to chemistry and deposition (with decay rate $\tau$, and thus the decay being $\sim \exp[-y/(\tau \cdot s)]$ and $\lambda_1 = \lambda/s$, equation B5). Equation B2, $f(x, y)$, describes the diffusion of our species perpendicular to the downwind direction.




## Appendix C: Implementation of the Point Source Method

The point source method involves a non-linear fit of OMI VCDs to the EMG as detailed in appendix B. Here, the solution is obtained using the non-linear least-squares solver "lsqnonlin.m" in Matlab. The non-linear terms are the lifetime and width parameters, whereas the mass is a linear scaling of the EMG.

It is important to note that the reference location must be specified as part of the point source method, where the reference location is the location of the emission source. In the case of a somewhat extended, or pseudo-point source, the centre of the emission source(s) is used. Here, wind information comes from the ECMWF ERA-interim or ERA-5 reanalyses (Dee et al., 2011).

In order to combine many overpasses where wind direction is constantly changing, a wind rotation scheme is used (Pommier et al., 2013) where the location of the OMI pixel (as denoted by the pixel centre) is rotated about the reference location so that wind directions are aligned (see also Fioletov et al. (2015)).

Background levels of $NO_2$ must be accounted for when deriving $NO_x$ emissions, where background in this context refers to the $NO_2$ that would be present if the source in question were removed. This can be done by additionally including a constant as part of the EMG fit, or, as here, an upwind average can be determined and subtracted from all VCDs before the fit. This is illustrated in Figure B1, which shows a sample EMG plume and the two domains used for the fit. The average over the upwind box (between $x = \pm 50$ and $y = -80$ to $-40$ km) is used to calculate the background, which accounts for any $NO_2$ not emitted locally, as well as potential offsets due to an imperfect stratospheric removal. The larger box is the domain over which the non-linear fit is performed ($x = \pm 40$ km and $y = -40$ to $120$ km).

While the fitting is carried out in the rotated frame, the agreement with the orignal mean OMI VCD dsitribution can be assessed by taking the fitted EMG and rotating it back to the original co-ordinates. This is called the reconstruction, and it highlights one advantage of using this 2D approach where spatial information is not lost in an averaging.

## Appendix D: Implementation of the Multi-source Method

To determine emissions from multiple sources, a related method has been developed (Fioletov et al., 2017; McLinden et al., 2020) whereby each (potential) source location is treated as an EMG plume. The total VCD is therefore the sum over all plumes, plus the background term,

$$\text{VCD}(x, y, s) = a_0 + \sum_{i=1}^{N} a_i \cdot EMG(x, y, s) \tag{D1}$$

where $N$ is the total number of sources. This is solved as a system of linear equations and thus the non-linear co-efficients in the Gaussian terms, $f$ and $g$, equations (B2) and (B3), are specified and not fit. The matlab function lsqlin.m was used to with a positive constraint placed on all values of $a_i$.

In this work source locations were specified as either being on a grid or at the centre of each surface mine.



**Table E1.** Summary of emissions calculations employing algorithm variants. Emissions given here are averages over the entire 2005–2022 period. Average NPRI stack emissions (2005–2022) are 30.5 kt[$NO_2$/yr]. The mean over all methods, and the variability among methods, is given in the bottom row.

| Emission run | Algorithm | $(\tau, \sigma)$ | Gridded Emissions | Point Sources Specified | Wind height adjustment | OMI Emissions kt[$NO_2$/yr] Fleet | Total |
|---|---|---|---|---|---|---|---|
| TOTAL1 | Point | Fitted | N/A | N/A | No | N/A | 78.0 |
| TOTAL2 | Multi | Specified | Yes | No | No | N/A | 74.0 |
| TOTAL3 | Multi | Specified | No | No | No | N/A | 72.4 |
| FLEET1 | Multi | Specified | Yes | Yes | No | 43.6 | 74.7 |
| FLEET2 | Multi | Specified | No | Yes | No | 43.6 | 74.7 |
| FLEET3 | Multi | Specified | Yes | Yes | Yes | 47.2 | 78.3 |
| FLEET4 | Multi | Specified | No | Yes | Yes | 42.2 | 73.3 |
| | | | | | *Mean ± Standard Deviation* | $44.1 \pm 2.1$ | $75.1 \pm 2.3$ |

## Appendix E: Emissions Calculations

Due to variations in solar zenith angle and cloud cover OMI does not sample evenly through the year, and any intra-annual changes in emissions may not be totally captured. To estimate this effect, monthly totals of bitumen mines (Alberta Energy Regulator, 2021a), which should be a good proxy for $NO_x$ emissions, were examined. From Figure E1, monthly values fluctuate about the annual mean by roughly 10–15%.

A correction factor was calculated by taking the ratio of annual bitumen mined using monthly values weighted with the OMI sampling to the annual total. This value varies from 1.00 to 1.05. It was then used to scale the retrieved emissions in Figures 9 and 10, and Table E1, below. These values are smaller than those derived for OMI $SO_2$ (McLinden et al., 2020) as coverage for $NO_2$ is less seasonally-dependent.

As discussed in section 2.4.3, the missing component of $NO_x$, NO, is accounted for using an externally-derived $NO_2/NO_x$ ratio. Figure E2 shows a comparison of observed ratios made during the 2018 aircraft campaign in which plumes were tracked downwind of the surface mines. An evolution is observed from values around 0.6 to 0.75 due to photochemical processing. These observations can be compared to the emissions-based method of estimating $NO_2/NO_x$, which accounts for downwind effect, appropriate for annual and June-July time periods, with the latter done to be more consistent with the observations time period. The good overall consistency suggests the emissions-based approach is reasonable. Further, an estimate of uncertainty is derived by looking at the variability.

Emissions calculations were carried out using the point source and several different variants of the multi source methods. The purpose was to test different assumptions as well as to conduct a sensitivity study that could be used to help quantify uncertainties. A summary is of the various emission runs is provided in Table E1.





Seven runs were used: the point source method, and six multi-source variants. Variables changed for the multi-source runs included: fitting all emissions using the gridded and mine-specific approach, fitting only fleet emissions (with point source emissions specified) using the gridded and mine-specific approach, and fitting only fleet emissions using the gridded and mine-specific approach but using higher altitude winds for the stack emissions and lower altitude winds for the fleet. As can be seen

from Table E1 there is little difference among them.

**Appendix F:  Resolution and the width parameter, $\sigma$**

The relationship between $\sigma$, $\sigma_{\mathrm{diff}}$, $\sigma_{\mathrm{size}}$, and $\sigma_{\mathrm{pixel}}$ was explored in appendix F using a simple model in which a collection of true-Gaussian plume point sources were used to simulate a near point source (of some radius $\sigma_{\mathrm{size}}$) and smoothed to satellite resolution (for pixel size $\sigma_{\mathrm{pixel}}$). Fitting the combined distribution to an EMG allowed a relationship between these various

terms to be established. Figure F1 shows $\sigma$ has has a behaviour reminiscent of $\sigma_{\mathrm{size}}^2$ and $\sigma_{\mathrm{pixel}}^2$, and for small values of both, values less than 1 km which suggests $\sigma_{\mathrm{diff}}$ is small. Figure F1 will be used below to help interpret results.

The behaviour of the fitted width parameter, $\sigma$, is further examined here in an attempt to understand how it varies with the spatial size or extent of the source, $\sigma_{\mathrm{size}}$, satellite pixel size, $\sigma_{\mathrm{pixel}}$, and the plume diffusion term, $\sigma_{\mathrm{diff}}$. To do this, a set of numerical experiments were performed in which a collection of true-Gaussian plume point sources were used to simulate a

near point source (of some radius $\sigma_{\mathrm{size}}$) and smoothed to satellite resolution (for pixel size $\sigma_{\mathrm{pixel}}$). The Gaussian plume is of the form given in Stockie (2011) (with $\alpha = 0.33$ km, $\beta = 0.86$) but integrated in the vertical to simulate VCDs. Each Gaussian was sampled at high spatial resolution, 0.1 km, and the composite distribution was calculated by summing over all Gaussian plumes. This was smoothed to coarser resolutions in order to simulate observations from satellite instruments.

An EMG was then fit to the resultant distribution in order to derive $\sigma$. By varying $\sigma_{\mathrm{size}}$ and $\sigma_{\mathrm{pixel}}$, Figure F1 is derived,

which shows a behaviour consistent with equation 1. For the smallest case explored, $\sigma_{\mathrm{size}} = 0.1$ km and $\sigma_{\mathrm{pixel}} = 0.4$ km, a $\sigma$ of 0.35 km was found. This can be interpreted as representing the diffusion term and confirms, at least for the EMG formulation, that it is small on this scale. Given this, the form,

$$\sigma^2 = a + b \cdot \sigma_{\mathrm{pixel}}^2 + c \cdot \sigma_{\mathrm{size}}^2, \tag{F1}$$

with $a = 0.33$ km$^2$, $b = 0.41$, and $c = 0.28$, is a good representation for the domain considered here. Interestingly, as $\sigma_{\mathrm{pixel}}$

was defined as a length of a square and $\sigma_{\mathrm{size}}$ as a radius, their weights are roughly equal when area is considered.

According to Figure F1 and with the effective pixel size of OMI (as defined as the square root of the average pixel area) being about 20 km, a $\sigma$ of 15 km is expected. From section 3.2, OMI views a point source with $\sigma = 11$ km, a value close but slightly smaller than the 13 km from this numerical experiment. The difference could be related to the fact OMI pixels are not square (as assumed here) and also there is a considerable amount of oversampling by using multiple years of observations.

In the context of this multi-source method, an estimate of the minimum separation distance required to resolve between neighbouring emissions can be estimated by considering their basis functions. The criteria adopted here is that emissions from neighbouring sources can found as long as the correlation between their basis functions does not exceed 0.5. Using this, the



minimum distance between sources as a function of lifetime, $\tau$, and width parameter, $\sigma$, is shown in Figure F2. For OMI, this suggests sources $\sim$22 km apart can be resolved. If a more relaxed threshold of 0.7 is used then this decreases to about 15.5 km.

Wind is also important: if the winds are halved there is less overlap of the plumes, and 20 km is required; likewise, doubling the wind speed increases this to 24 km. Note that while this suggests low wind speeds are advantageous, in reality this also means the plumes are not as evident making an accurate emissions calculation more difficult.

TROPOMI, with its significantly improved spatial resolution relative to OMI (a 12-fold improvement in area, or 3–4 fold improvement in distance), will mean much smaller values for $\sigma$, and hence an improved ability to resolve emissions.

Even though emissions from the multi-source method were simply summed to a total over all surface mining based on a minimum separation distance of OMI of about 22 km, it is nonetheless worth attempt to resolve them. Here we consider the approach in which one source is assigned to each mine (e.g., see Figure 8) and consider the period 2010–2020 where reported emissions are available for both stack and fleet sources. A scatterplot of the facility-specific emissions is shown in Figure F3

For distances between mines of less than about 12 km, with one exception, OMI has difficulty distinguishing between

neighbouring sources and can misassign emissions. The two larger emission sources, Syncrude-Mildred Lake and Suncor, each with reported emissions of about 20 kt[$NO_x$]/yr, are resolved fairly well. Even though Suncor is only about 7 km from its nearest neighbour, it also emits an order of magnitude more $NO_x$. Overall OMI does show some limited ability to resolve emissions between neighbouring mines. This may suggest that a correlation threshold of 0.5 is too stringent, and that perhaps a value of 0.7, which leads to a minimum separation distance of 17 km, is more reasonable.

*Author contributions.* CM wrote the paper and performed the analysis. VF, CM, DG, and ED developed the methodology. JZ and CA provided emissions and proxy-data . NK and LL provided the OMI data. All co-authors provided comments and feedback to CM.

*Competing interests.* There are not competing interests.

*Acknowledgements.* The authors would like to thank the Wood Buffalo Environmental Association (WBEA) for the provision of their in situ data. We acknowledge the NASA Earth Science Division for funding of OMI $NO_2$ product development and analysis, and the Air Quality

Research Division support teams and the National Research Council aircraft pilots and technical support team for the aircraft measurement campaign. These measurements were carried out as part of the Oil Sands 2013 and 2018 aircraft measurement campaign projects, funded by the Oil Sands Monitoring (OSM) program by the Governments of Alberta and Canada.



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





**Figure D1.** Three year mean OMI NO$_2$ vertical column densitiy (VCD) distributions over the surface mining. An averaging radius of 12 km was used. The OMI-derived total NO$_x$ emissions are indicated.





**Figure D2.** Three year OMI NO$_2$ vertical column densitiy (VCD) fit reconstructions over the surface mining, corresponding to the maps in Figure D1. An averaging radius of 12 km was used. The OMI-derived total NO$_x$ emissions are indicated.





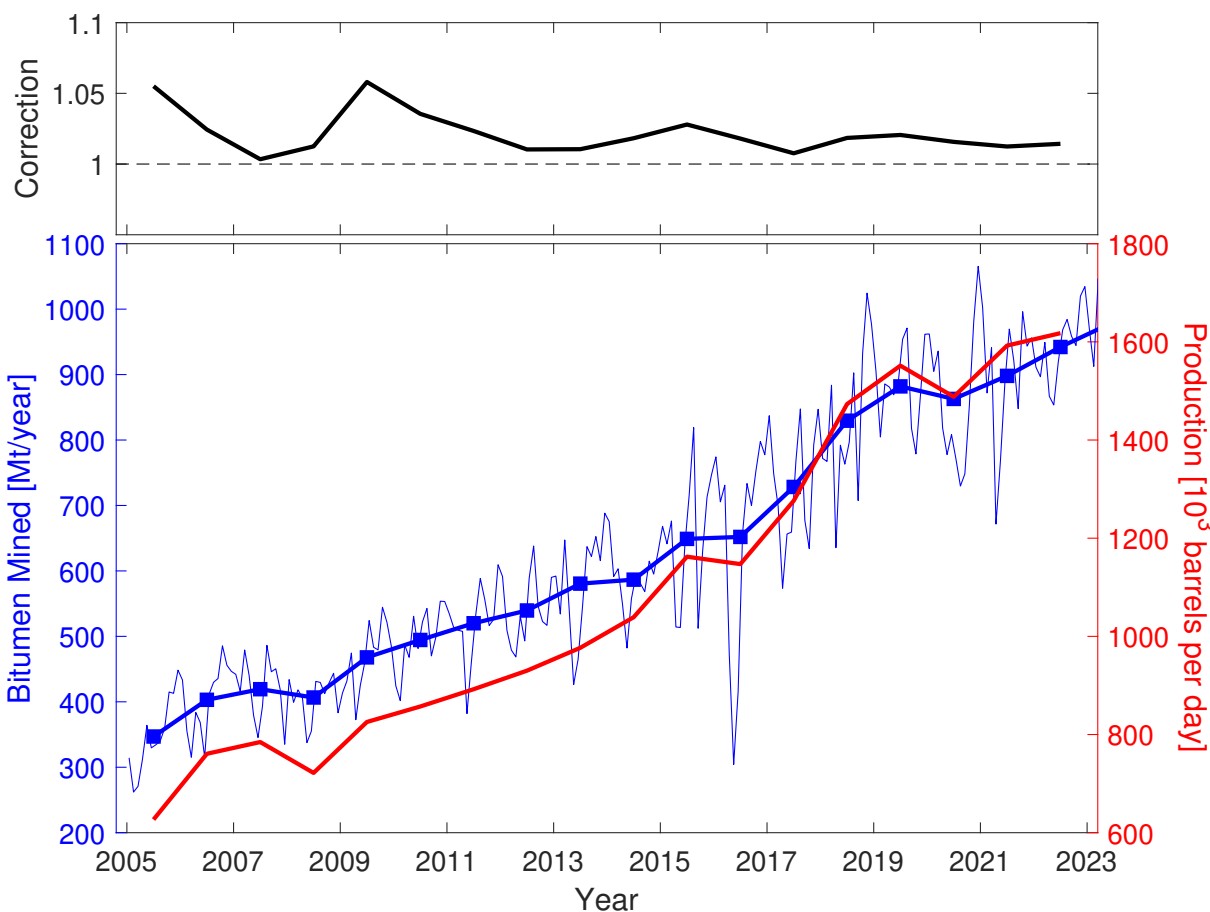

**Figure E1.** Total monthly and annual bitumen mined from the the surface mining region, and total production of synthetic crude from the surface mines. (The monthly bitumen mined has been converted to annual rates.) The panel at the top shows the correction factor applied to the annual emissions to account for the monthly sampling.



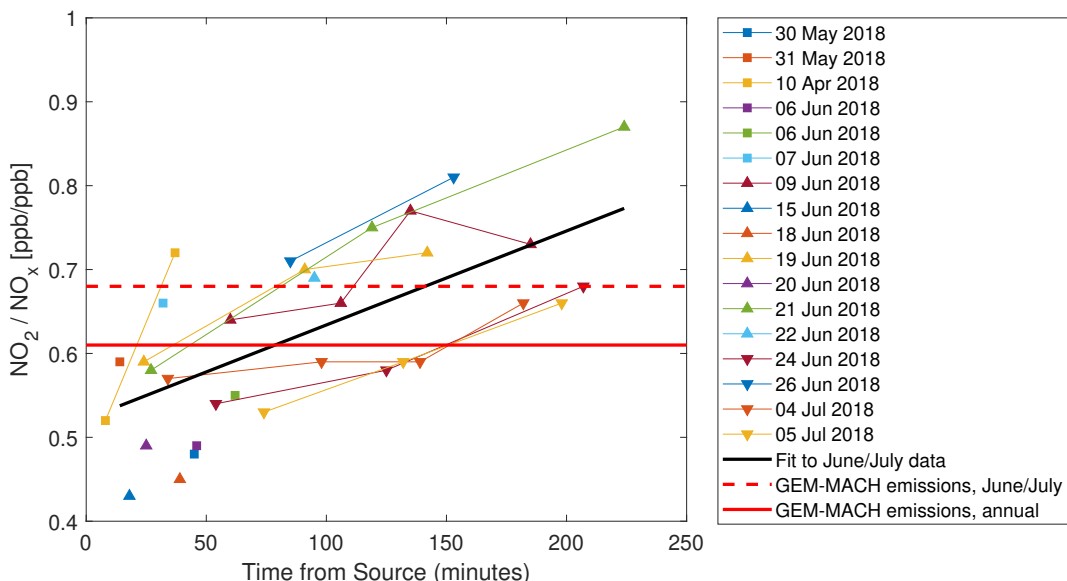

**Figure E2.** Comparison of NO$_2$–NO$_x$ ratios. Data from 8 flights in June and July 2018 taken during the oil sands aircraft campaign as a function of time downwind of the source, along with a fit. For comparisons, the red lines show the values using the emissions-based approach (see section XX) based on GEM-MACH model simulated OMI data. The standard deviation of the aircraft data about the line of best fit is 0.07, or 11%.





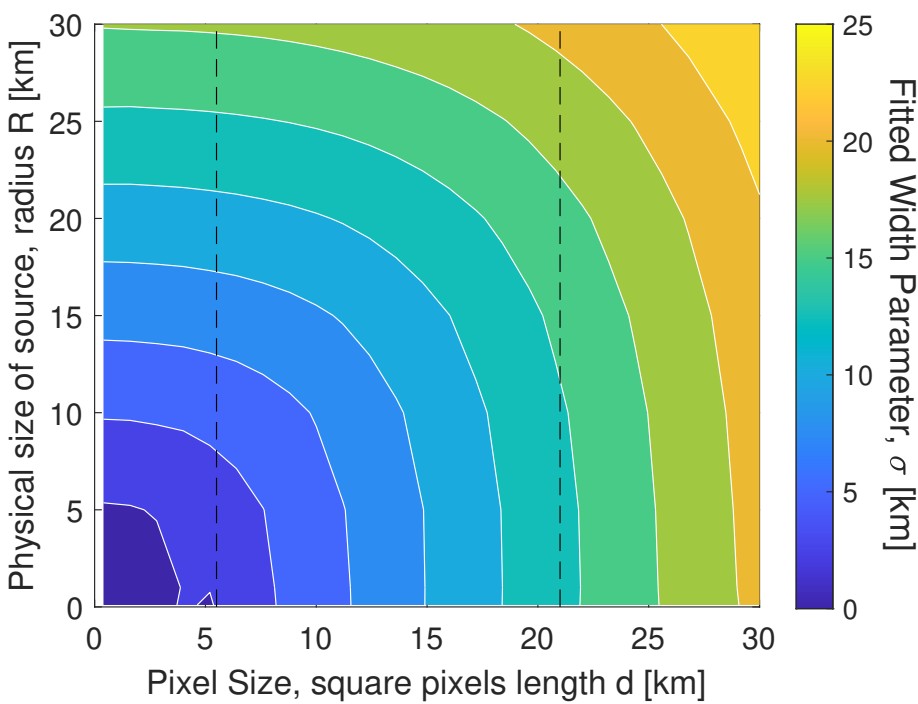

**Figure F1.** The relationship between the EMG width parameter, $\sigma$, and the physical extent of the emissions source and the satellite spatial resolution. See text for information on how this figure was derived. The vertical lines correspond to effective pixel sizes for the OMI and TROPOMI instruments, at 21 and 5.5 km, respectively.



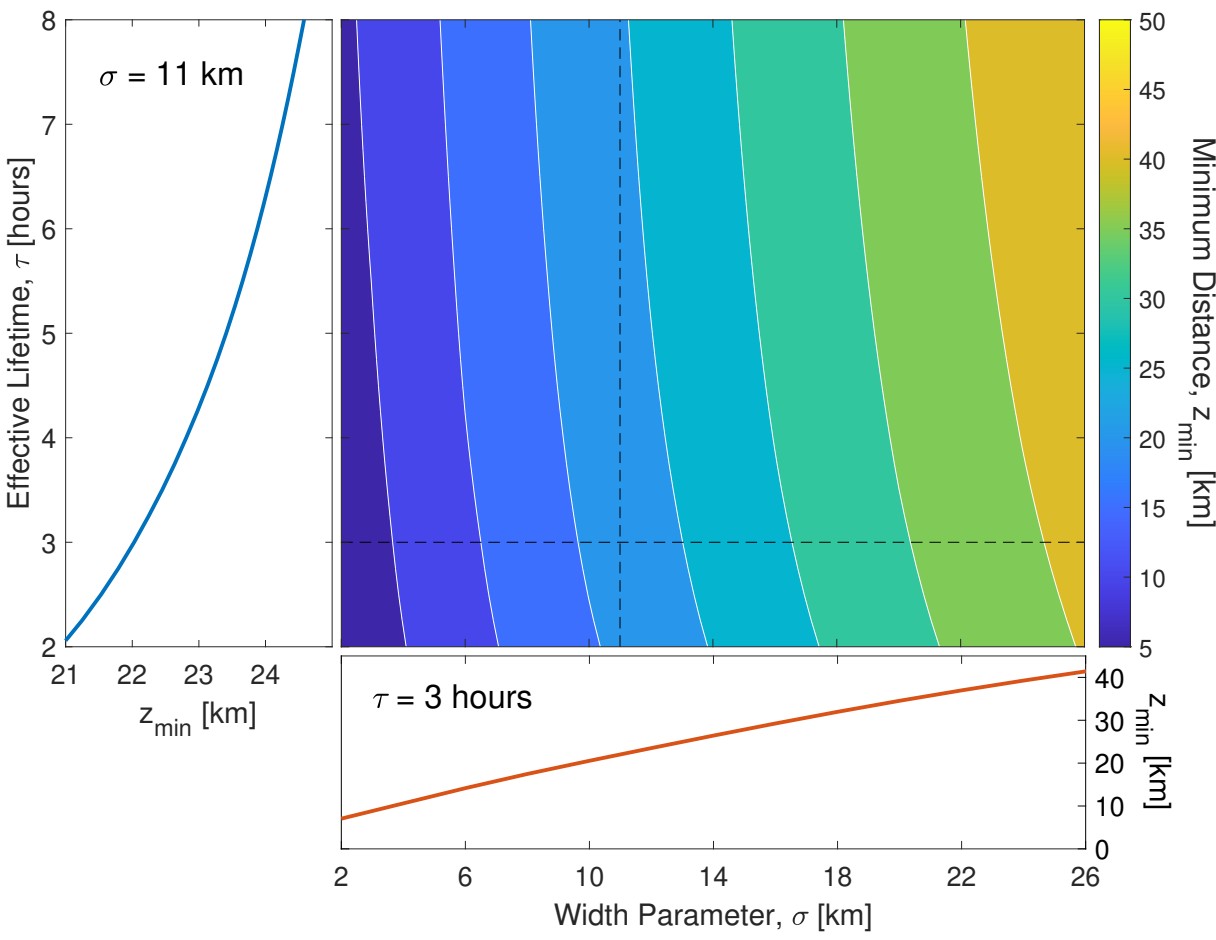

**Figure F2.** Minimum distance required to distinguish between two point sources as a function of plume width and effective lifetime. The line plots to the left and below show this quantity for $\sigma = 11$ km and $\tau = 3$ hours, respectively, and are cross-sections indicated by the white dashed line.





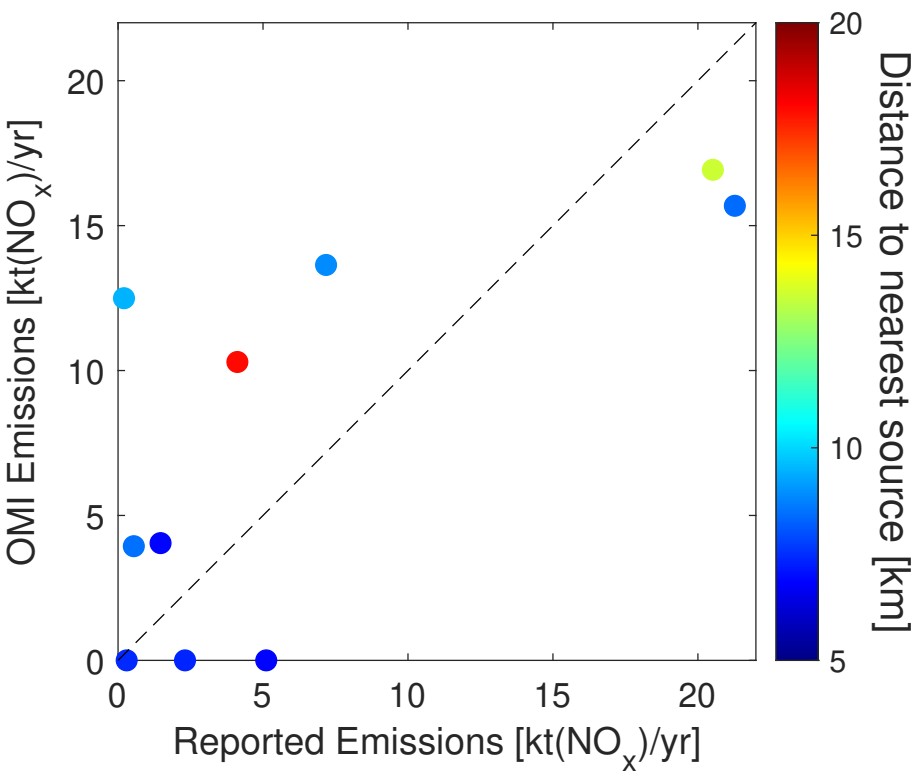

**Figure F3.** Comparison of reported and OMI-derived NO$_x$ emissions derived for individual mining operations considering the years 2010–2020, colour-coded according to proximity the their nearest neighbour.