# Peer review of "Monitoring of total and off-road $NO_x$ emissions from Canadian oil sands surface mining using the Ozone Monitoring Instrument"

_EGUsphere, 2024_

## Author Comment (AC1)

The authors thank reviewer 2 for their efforts in reviewing our manuscript. Our replies are in blue.

This study applies EMG approach to estimating NOx emissions from oil sands in Alberta, Canada. The authors further derived NOx emissions from large off-road mining fleets. Overall, the manuscript is well structured, and the results are robust with clear discussions of the uncertainties of the methods. I think this manuscript is almost ready for publication. I only have a few minor comments.

1. Line 565: It's not clear to me how the EMG method applies to multiple sources. My understanding is that the x,y,s are specific to each plume, but it's unclear to me how the x,y,s are defined with respect to multiple source locations. As the authors mentioned, one source location may affect the other, especially when they are nearby. It's unclear to me how the proposed methods could separate the influence from nearby sources. Maybe it'd be better if the authors could present a figure of several example plumes to explain the methods.

   We have added a figure as requested, and overall expanded on the existing explanation. As part of this we explain how x,y,s are specific to each plume (or location). This material was placed in Appendix D, which describes the multi-source method, and which we invite the reviewer to read. Not that ee also adjusted equation D1 to explicitly show (using script-"R") that the EMG is rotated back into the original lat-lon co-ordinates before being used:

$$\text{VCD}(x,y,s) = a_0 + \sum_{i=1}^{N} a_i \cdot \mathcal{R}[EMG(x,y,s)]$$

   As to the issue of separation, this was examined in detail in Figure 7 which summarizes how far apart two source need to be in order to be resolved (in section 3.2 and appendix F). The finding was that OMI is not generally capable of distinguishing between individual mines, and so this is why only the total (over the surface mining area) emissions are reported.

   Figure 2: Since the trend analysis is based annual emissions, I'd suggest the authors present a figure for a single year. Same for other figures. I think the novel part of this manuscript is the long-term trend, but the figures presented are mostly for multi-year average. It'd be useful if the authors could show the contrast between 2005 and 2022 to highlight the changes occurred.

   We have this figure (designated Figure C1) and supporting text to Appendix C which discusses additional detail of the method used. The text added read:

   Figure 2 shows the mean OMI VCD, OMI rotated VCD, fitted VCD, and reconstructed VCD plots considering all years of OMI data. Analogous plots are shown here but considering only a single 3-year period, 2005–2007, in Figure C1. Each three year period considered, even those in later years with reduced data, are comparable in fit quality. This time period was chosen as it predates much of the expansion of the northern mines, and thus better represents a point source as reflected by a smaller width parameter, $\sigma$ = 17.5 km vs 20.6 km, and smaller total emissions, E = 55 vs 70 kt(NO2)/yr, as compared to the all-year analysis.

2. Line 400: The emissions are reconstructed from NPRI emissions averaged from 2005 to 2020, but it seems that NPRI emissions vary yearly. How would this affect the derived trends?

First off, this was suppose to say 2005-2023 (the entire range of NPRI available that overlaps the OMI timeseries).  This text discusses the initial proof of concept, and is meant to be illustrative. Later in the paper this procedure was repeated for each year (or 3 year running window) and here NPRI emissions  - which do vary somewhat (as the reviewer points outs) – were used.

3.  Figure 3: Lifetime should be hours, not years.

Corrected.

---

## Author Comment (AC2)

Many thanks to review 1 for their detailed comments which have led to noticeable improvements in the manuscript. Author response are in blue.

Review of " Monitoring of total and off-road NOx emissions from Canadian oil sands surface mining using the Ozone Monitoring Instrument"

Using OMI observations, this study quantifies the point and area NOx emissions from the oil sands mining and fleet in Canada with two existing methods based on exponentially-modified Gaussian (EMG) plume model. The paper is well constructed and comprehensive. I recommend it to be published after adding some more details about the methods used including also its limitations.

We have addressed the specific comments below, and assume these cover what is meant by "more details about the methods used including also its limitations"

Detailed list of comments

Line 30: Consider adding CH4 in the list of pollutants (see https://doi.org/10.1029/2021GL094151)

Thanks. This was added. And we also added a CH4 paper by Daniel Varon et al. as well.

Figure 1: (1) the dark blue lines are difficult to distinguish from the dark background. Consider using "light blue".
Changed, as requested.
(2) Please, add the locations of Fort Mckay and Fort Chipewtan on the map.

Fort McKay was added (actually called Bertha Ganter – Fort McKay now so this was changed). Fort Chip is too far north to show on this map, but its location is now described.

Line 89-90: Different units are used for the emissions, which makes it difficult to compare. Please, use one type of unit throughout the paper.

These are typos, and all instances should have read "kt[NO2/yr]". This was corrected.

Line 122-126: For the OMI instrument the row anomaly plays an important role. Especially for the trend analysis, it is important to know what the number of observations are that are used for each year. In addition, in order to understand how representative the measurements are I would like to know what are the number of observations per month that are used in the annual averages (or 3-year averages). Please, add this information to the paper.

This information was added. Note that a seasonal correction was already performed in order to help eliminate any possible bias due to an uneven seasonal sampling (Figure E1 in original, Figure E2 in revised manuscript). The correction is based on monthly production numbers, which is considered to be a good proxy for NOx emissions.

A figure showing the annual and monthly data density and an accompanying paragraph were added in Appendix E:

Over the course of the OMI mission, a blockage known as the row-anomaly has meant a (generally) expanding number of pixels become unreliable, leading to a reduction in data density. It is for this reason that data are analyzed in three-year periods as opposed to 1 year. Figure E1a shows the impact of the row anomaly on observations within 80 km of the reference location (57.1∘N, 111.6∘W). Due to

variations in solar zenith angle and cloud cover OMI does not sample evenly through the year. This variation is illustrated in Figure E1b, with each year generally following this same average pattern. There is uneven sampling throughout the year with little data in November- January due to low sun angles, and a drop in the summer due to increased cloud cover.

[Figure]

The row anomaly is also affecting the pixels in nadir-viewing, and therefore the resolution is changing during the time series, which is also worthwhile to discuss here.

Agreed. The following text was added here: "Note that pixels affected by the row anomaly change over the course of the mission which impacts both the data density and, as discussed below in Figure 3c, the effective spatial resolution. See Torres et al. (2018) for more information on the evolution of the row anomaly"

Line 138: What is the reason of partly using ERA-Interim instead of using ERA-5 for the whole time period?

This is a mistake, a text artifact from a previous data version. ERA5 is used throughout now. This was corrected.

Line 158: This reference to equation (B4) in the appendix, makes it hard to read this analysis. Please include equation (B4) in the main text.

We agree that it would be convenient to have this equation here, but on the other hand, eq (B4) is part of a series of equations and moving it would mean (B1)-(B5) should all be moved. In the balance, we feel the equations, plus the requires associated text, in the main paper would be too much of an aside (in a paper that perhaps too many already). As such we respectfully keep the equation where it is.

Line 166: Figure F1 is often referred to in the text. Why not place in the main text?

That is a fair point. Similar to the point above, while it could be moved easily have been, since there are already 12 figures in the main paper which seems like a lot to begin with we elected not to move it. (In this paper we wanted to do a deeper dive into these emissions methods and so there are multiple side studies trying to better understand them, and documenting these as we tried to do means having many plots in an appendix.)

Line 181: The "single, average plume" is an composition of plumes with various wind speeds. What does this mean for the definition of the wind speed s in equation (B1)?

Equation (B1) uses individual wind speeds, matched in space and time to each OMI pixel. The average plume is then based on a distribution of these individual wind speeds. One can also examine how the average plume changes by limiting the analysis to, e.g., stronger winds (smaller peak VCD values but longer tail) or weaker ones (higher peak values, less pronounced plume shape).

Section 2.4.1. Important feature of this method is that it assumes a known location of the sources. I think this is important to mention here.

We tried two approaches: one where we assumed the location of the source (sample results shown in Figure 7), and one where we did not. In the case where we did not assume this, we allowed for an 8 km x 8 km grid of potential source locations over the domain being considered (Figure 6). This text was added near the beginning of section 3.2 to make this clearer:

"If the locations of the emissions are well known, then these can simply be specified; alternatively, a grid of potential source locations can be used. In this work both options were explored."

Section 2.4.2 Here the assumption is made that the lifetime is constant over the whole region and independent on the strength of source. In strong sources the lifetime can become longer inside the plume (due to OH depletion).

This is a good point. Lifetime could vary from source to source depending their individual emissions rates. It may be possible to parameterize this dependency, initially assuming a single lifetime for all sources to derive emissions as done here, and then adjusting lifetimes accordingly, perhaps iterating this a couple times to update the emissions rate. Adding this here is beyond the scope of the present study. Further, we argue that as we are only reporting on the *sum* of surface mining emissions, there will be some cancelation of errors. Had we attempted to report on individual emissions this could be much more important.

As it stands, we added the following text:

"Here $\tau$ is taken from the point-source analysis in Figure 3(b) and used for all individual sources. This is a simplification as it was already demonstrated in Figure 4 that $\tau$ depends on VCD, and hence local emissions. It is argued here that this will be a secondary effect since (i) plumes from individual sources will frequently overlap due their proximity and the coarse spatial resolution of OMI, and (ii) significant cancellation of errors since the point-source lifetime used was for the combination of all local sources, small and large. In the future, when isolating individual sources is the goal, it would be worthwhile to explore a parametrization which might account for this dependency."

The lifetime is also an average of the whole year, while the number of observations are changing over

the year (less observations in the winter time) and change from year-to-year. I wonder how this affect the uncertainty of the lifetime used in this method. Please, add more discussion to the text.

We added how the observations per year, and per month, in a new Figure, E1. The monthly profile is consistent from year to year, and so each year the lifetime will be biased towards the non-winter months. But the lifetime reflects the data available and only introduces a bias if there is some seasonal pattern in emissions, which does not appear to be the case based on Figure E2. The decline in observations with time from the row anomaly will add uncertainty as there are fewer data points in the fit. This can be seen in Figure 3b as the errors bars roughly double over the course of the mission. However compared to other sources of error like the uncertainty in wind speed, this does not substantially impact the overall uncertainty in emissions.

In the discussion for Figure 3 there is the statement:

"The increase in the 1-sigma fitting error bars (in all [Figure 3] panels, a--c) reflects the decrease in the number of OMI pixels due to the onset and expansion of the row anomaly."

Line 213: "Observations of of ". One "of" too many.

Corrected

Line 214: How can the authors draw this conclusion that the plume resides between100 and 800 m, if the altitudes below 100 m are not sampled?

We assume that the plumes extend down to the surface. Line 214 states "… plumes were found to reside at altitudes between the lowest altitudes sampled by the aircraft (100–300 m above the surface) and 800 m" which merely summarized the aircraft observations. We go on to state (line 216) that we assume the plumes reside between the surface and 800 m. We note here that we assumed the plume extends below 100 m (despite not sampling these heights) as the plumes at the lowest height sampled were still substantial and must have extended to some height below. Whether a plume went to the surface or 50 m, for example, is not important when compared with other uncertainties.

We modified the beginning of the sentence in question to "These aircraft data indicated that NOx within plumes were found to reside at altitudes between …" in order to reinforce that it is the aircraft data being discussed.

Line 220: "the the oil sand", one "the" should be removed.

Corrected

Line 225-235: this NOx/NO2 ratio seems also dependent on the season and location, so I wonder if the 8% uncertainty is maybe an underestimation. Can you specify what the variation is over the year based on the GEM-MACH model ?

This is not a straightforward quantity to try and arrive at an uncertainty for. We looked at the monthly variability in this quantity using the GEM-MACH model as the reviewer suggested and found it varied between 1.5 and 1.8, or about ±9% about the mean. However, the actual value used, 1.63, was based on GEM-MACH output, sampled according to OMI (which accounts for the seasonality in the number of observations used), and also accounts for the change in this ratio downwind of the source. Thus the seasonal cycle itself is not that relevant when estimating uncertainties as this is folded into the 1.63

value used. In the end we chose the variability from the summertime aircraft observations as an uncertainty since there appeared to be consistency between it and the values derived from the GEM-MACH model.

Figure 2: (1) A discrete number of colors is shown in the Figure, while the legend shows a continuous color bar. Please adapt this, also for the other Figures in the manuscript with similar mismatch between Figure and legend.

We have changed the colorbars of all figures to discrete levels, as suggested.

(2) I assume the triangle is at (0,0) for the rotated VCD figures. It will help if the triangle is shown at this location.

This was added.

(3) In Figure (c), showing the reconstructed VCD, the distribution looks very symmetrical. Why does it not look more like Figure (a) with dominating winds from the South ?

Figure 2(a), which shows the mean observed NO2 VCD, is elongated in the N-S direction primary due to the distribution of emissions from the various mining operations (see Figure 1). It is true that this is exaggerated due to the distribution of wind speed and direction, with winds out of the W and SW the most common.

Figure 2(c) is largely symmetrical due to the fact that in this simpler, assumed point-source, emissions calculation all emissions are assumed to originate from the reference location, given by the triangle. As such the reconstruction in (c) will only depart from a symmetrical pattern to the extent the wind directions, or wind direction and speed, have a preferred direction.

Some of the discussion above was used to elaborate on these panels in the revised manuscript. Please see line 295

Figure 3: (1) The figure captions are often difficult to understand without reading the text. For example, in Figure 3, it is not clear what the "VMR: and "Effective pixel size" mean.

These were clarified. VMR is not defined in the text and figure caption. And the definition of effective pixel size is now given in the figure caption.

(2) The exceptional year 2005 is explained in the caption. What about 2022? Is that also for two years?

No, only 2005. Note that we have added one additional year to the data record and so 2023 (using 2022-2024 data) is now the last time period considered. We have gone over the paper to make sure this is the consistent message.

Line 317-318: A large lifetime variation has been found within the individual NOx plume from Krol et al. (2024). Therefore, it might be useful also to refer to this more recent paper: https://doi.org/10.5194/acp-24-8243-2024.

This is a nice paper, and their findings are in line with those here (re/ NOx:NO2 ratio and lifetime). It is now mentioned in a couple locations in the revised manuscript.

Line 344: Change the Roman "*tau*" into the Greek τ.

This was corrected.

Line 376: How was the 22 km derived. Can you add a reference?

The sentence quoting a value of 22km was a little out of order.  The 22 km came from an analysis described in appendix F.  By rearranging this section it should be much clearer now.

Figure 7. Results are shown for species SO2 and NH3, which are not discussed in the paper. I suggest to remove these points.

These NH3 and SO2 points are shown as other examples, similar to the TROPOMI and TEMPO points; no data from any of these are used in this work.  The methods in this paper are general and can (and have) been applied to other instruments and/or other species (such as SO2 from the OMI), and in that sense the findings go beyond our application to NOx in the oil sands. We argue that these examples would be of interest to others, and that since all we have done is simply plot points according to documented instrument spatial resolution and the effective lifetime as found and published by others, it is reasonable to include them.

Figure 10: These trends may be affected by the different sampling of OMI due to the row anomaly. This may explain the changes around the year 2007/2008. For a trend analysis it would be better to use exactly the same sampling in all years, as if the row anomaly already existed in 2005. This would give a better trend estimate.

This is a good idea, and something we had done but not reported on.  We examined this and found very little difference in emissions.  In 2006 and 2007 emissions were about 2-3 kt/yr smaller using this additional filtering, roughly 3-4%, which meant a slightly smaller growth 2005-2007 and slightly larger 2007-2009.  All in all the effect was marginal, well below the single year precision, and so in the end we did not change our filtering but we did add text to section 3.3:

> As the time period over which the NOx emissions were found to increase (2007-2011) generally corresponds to the main expansion of the OMI row anomaly, emissions over the entire time series were recalculated but restricted only to track positions that were unaffected by the anomaly over the entirety of the mission to date.  Considering the previous criteria of excluding pixels near the edge of the detector due to their coarse spatial resolution, only track position between 8 and 23 were considered for this sensitivity study.  Emissions calculated in his way were found to be minimally impacted, reaching roughly 3 kt(NO2)/yr in 2007, well below the calculated emissions precision value, and not changing the overall timeseries picture.

Line 577: "A correction factor....": This sentence is difficult to understand. Please, explain better what steps you took here.

We agree this was not particularly clear.  This was re-written as:

> A correction factor to account for this unequal sampling throughout the year was used.  A monthly total bitumen-mined value was assigned to each OMI pixel (used in the emissions calculations), according to its month and year.  For each year, the ratio of the mean monthly

bitumen-mined to the average (over all OMI pixels) of the OMI-sampled bitumen-mined was computed.

Figure E2, Page 41: "See section XX". Specify the section number.

This was corrected to "section 2.4.3"